# A pivot-tether model for nucleosome recognition by the chromosomal passenger complex

Reinis R Ruza[1], Chyi Wei Chung[2,5], Danny B H Gold [1,5], Michela Serena[1,3], Emile Roberts [2,4], Ulrike Gruneberg [2] & Francis A Barr [1✉]

## Abstract

**Spatial restriction of Aurora B to T3-phosphorylated histone H3 (H3pT3) nucleosomes adjacent to centromeres during prometaphase and metaphase enables it to phosphorylate proteins necessary for spindle assembly checkpoint signalling and biorientation of chromosomes on the mitotic spindle. Aurora B binding to H3pT3-nucleosomes requires a multivalent targeting module, the chromosomal passenger complex (CPC), consisting of survivin, borealin, and INCENP. To shed light on how these components mediate CPC localisation during prometaphase and metaphase, we determined the structure of the CPC targeting module in complex with haspin-phosphorylated H3pT3-nucleosomes by cryo-electron microscopy. This structure shows how the N-terminus of borealin and the survivin BIR domain act as pivot and flexible tethering points, respectively, to increase CPC affinity for H3pT3 nucleosomes without limiting it to a specific orientation. We demonstrate that this flexible, yet constrained pivot-tether arrangement is important for the control of spindle assembly checkpoint signalling by Aurora B.**

**Keywords** Mitosis; Spindle Assembly Checkpoint; Centromeres; Aurora Kinase B; Nucleosomes
**Subject Categories** Cell Cycle; Structural Biology

## Introduction

Spatial and temporal control of the key cell cycle kinase Aurora B is crucial for mitotic chromosome condensation, chromosome biorientation on the mitotic spindle, as well as the kinetochore recruitment of the principal spindle assembly checkpoint kinase, MPS1 (Carmena et al, 2012; Hayward et al, 2019; Hayward et al, 2022; Saurin et al, 2011; Vader et al, 2006; van der Horst and Lens, 2014). Accordingly, the specific subcellular localisation of Aurora B to chromosome arms and inner centromeric or pericentromeric chromatin during mitosis is thought to be crucial for regulation of

these processes. Upon anaphase onset, Aurora B is transported from chromosomes to the anaphase central spindle by the type 6 kinesin motor protein MKLP2 where its function is important for cytokinesis (Gruneberg et al, 2004; Hill et al, 2000; Serena et al, 2020).

Aurora B activation and localisation require 3 additional proteins, the Inner Centromere Protein INCENP, survivin, and borealin, which together form the chromosomal passenger complex (CPC) (Carmena et al, 2012). This tetrameric protein complex contains an active kinase module formed by Aurora B and the C-terminal IN-box motif of INCENP (Segura-Pena et al, 2023; Sessa et al, 2005), and a targeting module consisting of survivin, borealin, and the N-terminus of INCENP (Jeyaprakash et al, 2007; Klein et al, 2006). Because the IN-box motif is located more than 800 residues away from the N-terminus of INCENP, there is a clear physical separation of these two modules, potentially important for chromatin tethered Aurora B to reach substrate proteins at kinetochores involved in checkpoint signalling, microtubule attachment and chromosome biorientation (Cheeseman et al, 2006; Cimini et al, 2006; DeLuca et al, 2006; Liu et al, 2009; Tanaka et al, 2002; Welburn et al, 2010).

In addition to the core CPC, Aurora B targeting to chromatin at different stages in mitosis requires accessory factors and landmark signals for pericentromeric chromatin. CPC is initially recruited to heterochromatin in late S-phase due to an interaction between heterochromatin protein 1 (HP1) and INCENP, thus enabling Aurora B to mediate chromosome condensation at very early stages of mitosis (Abe et al, 2016; Ainsztein et al, 1998; Nozawa et al, 2010). This interaction is attenuated during prophase when Aurora B phosphorylates histone H3 at S10 promoting dissociation of HP1 from chromosome arms (Fischle et al, 2005; Hirota et al, 2005). Hence, CPC association with chromosomes now becomes more dependent on histone modifications. In prometaphase and metaphase, the CPC remains associated with chromosome arms but becomes strongly enriched at centromeres due to two histone tail modifications, T3-phosphorylated H3 (H3pT3) and T120-phosphorylated histone H2A (H2ApT120), which are created by the kinases Haspin and Bub1, respectively (Dai et al, 2006; Wang et al, 2010; Wang et al, 2011; Yamagishi et al, 2010). Recent structural studies reveal that the haspin kinase domain binds directly to nucleosomal DNA in order to phosphorylate H3T3

[1]Department of Biochemistry, University of Oxford, South Parks Road, Oxford OX1 3QU, UK. [2]Department of Pathology, University of Oxford, South Parks Road, Oxford OX1 3RE, UK. [3]Present address: Arctoris Ltd, 120E Olympic Avenue Abingdon, OX14 4SA Oxfordshire, UK. [4]Present address: Biochemistry Department, University of Geneva, 30, Quai Ernest-Ansermet, 1205 Geneve, Switzerland. [5]These authors contributed equally: Chyi Wei Chung, Danny B H Gold. ✉E-mail: francis.barr@bioch.ox.ac.uk

(Hicks et al, 2025). H3pT3 is recognised by the baculoviral IAP repeat (BIR) domain of survivin, whereas H2ApT120 is thought to be recognised indirectly by borealin and survivin, via Shugoshin 1 (Sgo1) (Abad et al, 2022; Abad et al, 2019; Bonner et al, 2020; Du et al, 2012; Jeyaprakash et al, 2011; Kawashima et al, 2007; Kelly et al, 2010; Liu et al, 2014; Niedzialkowska et al, 2012; Serena et al, 2020; Tsukahara et al, 2010; Wang et al, 2010; Yamagishi et al, 2010). Binding of the N-terminal tail of Sgo1 to survivin has been proposed to mimic the interaction between H3pT3 and survivin, thus effectively resulting in two distinct binding modes of centromere-localised CPC (Abad et al, 2022; Abad et al, 2019). Both modes of CPC interaction with centromeric and/or pericentromeric chromatin, indirectly via Sgo1 and directly via H3pT3, respectively, have been reported to be important for both Aurora B-mediated correction of erroneous microtubule–kinetochore attachments during chromosome biorientation and for spindle assembly checkpoint signalling (Broad et al, 2020; De Antoni et al, 2012; Hadders et al, 2020; Wang et al, 2012). These interactions between the CPC and nucleosomes are thought to be crucial to retain Aurora B at the pericentromeric chromatin during spindle formation as microtubule-dependent pulling forces pull kinetochores apart during the chromosome biorientation process. This results in a large physical separation of Aurora B from the kinetochore at bioriented chromosomes reducing phosphorylation of proteins involved in error correction and spindle checkpoint signalling (Lampson and Cheeseman, 2011; Liu et al, 2009). Although the importance of spatial control of Aurora B activity in checkpoint signalling and chromosome biorientation is supported by a large body of data (Ditchfield et al, 2003; Hayward et al, 2019; Santaguida et al, 2011; Saurin et al, 2011; Vader et al, 2007), some evidence suggests that only Aurora B activity alone rather than its specific localisation is sufficient to trigger the spindle assembly checkpoint in the absence of Bub1 and Haspin activity (Hadders et al, 2020).

Supplementing the interactions mediated by nucleosome modifications with survivin, both INCENP and borealin contribute additional contacts that increase the affinity of the CPC for nucleosomes. Combined crosslinking and mass spectrometry studies have suggested that the borealin N-terminal region interacts with the nucleosome acidic patch, whereas its C-terminal loop regions also exhibit some affinity for DNA (Abad et al, 2019). A dimerization domain within the C-terminus of borealin and phase separation properties reported for full-length borealin might further contribute to targeting of CPC to nucleosomes by increasing the avidity for chromatin (Abad et al, 2019; Bekier et al, 2015; Bourhis et al, 2009; Trivedi et al, 2019; Trivedi and Stukenberg, 2020). Additionally, a positively charged RRKKRR motif present at the N-terminus of INCENP has been shown to promote chromosome binding through direct but low affinity interactions with DNA (Serena et al, 2020). Remodelling of these different interactions is critical for enabling the relocation of Aurora B from chromosomes to the central spindle in anaphase. Crucially, histone H3pT3 and INCENPpT59, close to the RRKKRR motif, are dephosphorylated at the onset of anaphase, allowing the CPC to associate with MKLP2, initiating transport away from chromatin to the central spindle (Gruneberg et al, 2004; Hummer and Mayer, 2009; Kitagawa et al, 2014; Qian et al, 2011).

The presence of several different nucleosome-recognising regions within the CPC implies that their actions must be coordinated to enable efficient binding to different sites on chromatin. Despite progress on understanding the individual interactions, there is nonetheless a lack of structural data for CPC bound to nucleosomes to guide functional studies. We therefore set out to determine the structure of the core CPC localisation module bound to the H3pT3-containing nucleosome and understand the relative importance of pericentromeric localisation of Aurora B compared to global Aurora B activity in promoting recruitment of the key spindle assembly checkpoint kinase MPS1 to kinetochores during mitotic spindle formation.

## Results

### Pericentromeric CPC is required for recruitment of MPS1 to kinetochores in prometaphase

To better understand the localisation of Aurora B and other CPC components we performed lifetime-STED microscopy which has the potential to discriminate different subregions of the centromere and kinetochore with a resolution of up to 50 nm. We focused on the difference between unaligned prometaphase chromosomes and bioriented chromosomes attached to the mitotic spindle. Kinetochore pairs, defined by NDC80-HaloTag, on unaligned prometaphase chromosomes were resolved as two peaks separated by $0.46 \pm 0.11\,\mu m$ (Fig. 1A, prometaphase—NDC80). Aurora B and survivin were seen as a broader, single peak which co-localised and overlapped with the twin peaks of the NDC80 kinetochore signal (Fig. 1A, prometaphase—AURKB and survivin). Bioriented metaphase chromosomes showed increased separation of the NDC80 kinetochore peaks to $1.37 \pm 0.22\,\mu m$ from $0.46 \pm 0.11\,\mu m$ in prometaphase (Fig. 1A, lines graphs and bar graph), with the Aurora B and survivin signals colocalised to a single broad peak resolved from the kinetochore (Fig. 1A, metaphase—AURKB and survivin). Like survivin, borealin colocalised with Aurora B in prometaphase and metaphase, discrete from the peaks of the kinetochore protein NDC80 which again showed increased separation of $1.62 \pm 0.16\,\mu m$ in metaphase from $0.58 \pm 0.22\,\mu m$ in prometaphase (Fig. EV1, prometaphase, metaphase—NDC80, AURKB, borealin).

To understand whether Aurora B and other CPC components were located at the centromere or pericentromere, we next explored the localisation of the centromere protein, CENPA, which forms the base of the NDC80-containing kinetochore structure. The CENPA signal was observed adjacent to the NDC80-HaloTag signal on both unaligned prometaphase and bioriented metaphase chromosomes (Fig. 1B, prometaphase, metaphase—NDC80, CENPA). Interestingly, prometaphase kinetochore pairs appeared to sit in a side-by-side arrangement that became extended in the metaphase configuration, so that CENPA was positioned on the inward side of the NDC80 signal (Fig. 1B, prometaphase, metaphase—NDC80, CENPA). The peak signals for CENPA and NDC80 were located within $0.10$–$0.15\,\mu m$ in both prometaphase and metaphase cells (Fig. 1B, prometaphase, metaphase). We therefore conclude that the CPC components Aurora B, borealin and survivin localise to pericentromeric chromatin which becomes extended during chromosome biorientation and physically distanced from the kinetochore, whereas the CENPA-defined centromere region and NDC80 kinetochore structure although

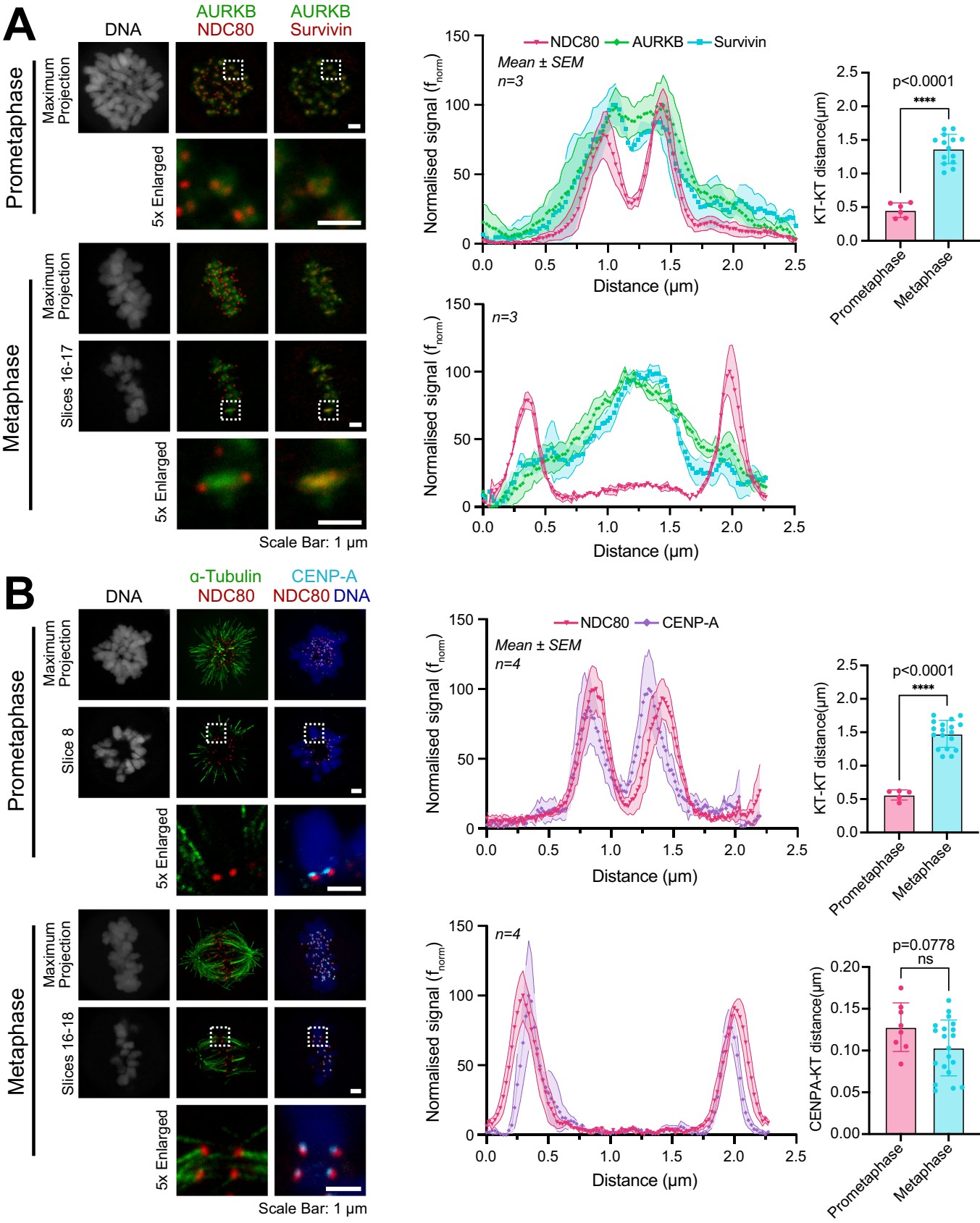

Scale Bar: 1 μm

◄ **Figure 1. The outer kinetochore and Aurora B-CPC are spatially separated in metaphase.**

(A) STED images depicting NDC80, survivin and AURKB, or (B) NDC80, CENP-A and α-tubulin localisation in prometaphase and metaphase HCT116 NDC80-HaloTag cells. DNA was stained with Picogreen. Representative maximum intensity projections and selected slices are shown with 1 μm scale bars. Line scans show signal intensity across kinetochore pairs (mean ± SEM, sample size $n = 3$ or $n = 4$ indicated in figure). Bar graphs show mean outer kinetochore-kinetochore (KT-KT) and outer KT-CENPA distances in prometaphase and metaphase (mean ± SD). For KT-KT distances in (A) prometaphase $n = 6$, metaphase $n = 14$ with a two-tailed unpaired $t$ test, $P < 0.0001$ (****). For KT-KT distances in (B) prometaphase $n = 5$, metaphase $n = 18$ with a two-tailed unpaired $t$ test, $P < 0.0001$ (****). For KT-CENPA distances in (B) prometaphase $n = 8$, metaphase $n = 20$ with a two-tailed unpaired $t$ test, $P = 0.0778$ (ns). In all panels, results are representative of over three independent experiments.

reoriented, undergoes no discernible change in separation. This is consistent with a model in which microtubule pulling forces physically move the kinetochore away from Aurora B during chromosome alignment (Lampson and Cheeseman, 2011; Liu et al, 2009). Sgo1 showed a different pattern of localisation to Aurora B and the CPC components with two peaks close to NDC80 which marks the outer kinetochore (Appendix Fig. S1A). Crucially, Aurora B remained at pericentromeres in Sgo1-depleted cells which undergo premature sister chromatid separation due to cohesion failure (Appendix Fig. S1B). Sgo1 therefore does not match the precise pattern of CPC localisation at pericentromeres in metaphase and is not essential for Aurora B recruitment under the conditions used here. We therefore focused on the core CPC subunits for further experiments.

As mentioned, Aurora B activity rather than its specific localisation has been proposed to trigger the spindle assembly checkpoint (Hadders et al, 2020; Hayward et al, 2019; Hayward et al, 2022; Santaguida et al, 2011; Saurin et al, 2011). To test whether Aurora B activity alone, or localisation and activity are important for the spindle assembly checkpoint we analysed the localisation of MPS1 in cells depleted for the Aurora B activation and targeting protein INCENP, or survivin or borealin which are required for Aurora B targeting but not directly for its activity. Depletion of either INCENP, borealin or survivin resulted, in all cases, in a near complete loss of Aurora B from pericentromeres and MPS1 from kinetochores in prometaphase-arrested cells (Fig. 2A–C; Appendix Fig. S2A, AURKB and MPS1), and reduced levels of the spindle checkpoint protein BUBR1 at kinetochores (Appendix Fig. S2B). INCENP depletion, due to its direct role in Aurora B activation resulted in a slightly greater reduction in histone H3 S10-phosphorylation (H3pS10) across the entire chromosome than depletion of either borealin or survivin (Fig. 2A,D, H3pS10). Depletion of borealin and survivin indirectly reduced INCENP stability as reported previously (Appendix Fig. S2C,D) (Gassmann et al, 2004; Honda et al, 2003; Klein et al, 2006), providing a plausible explanation of why H3pS10 is also reduced under these conditions (Fig. 2D). In summary, these data demonstrate that both normal CPC localisation as well as activity are prerequisites for efficient MPS1 recruitment to kinetochores. One caveat is the length of time taken to deplete the CPC subunits, and experiments using rapid acting chemical inhibitors in mitotic cells were therefore performed. Specific chemical inhibition of Aurora B for 10 min did not affect Aurora B localisation but prevented recruitment of MPS1 to kinetochores when the spindle checkpoint was reactivated with nocodazole in MG132-arrested metaphase cells (Fig. 2E–G, Control, AURKBi). By contrast, inhibition of the kinase Haspin which generates the T3-phosphorylation on histone H3 resulted in the decrease of centromere-enriched Aurora B and had a significant albeit less

pronounced effect on MPS1 localisation to kinetochores under the same conditions (Fig. 2E–G, Control, HASPKi). This may reflect the role of other pathways for CPC localisation, the relatively small, <fivefold difference in affinity of the CPC targeting module for T3-phosphorylated histone H3 versus unphosphorylated histone H3 determined in vitro (Abad et al, 2019; Serena et al, 2020), or only partial inhibition of Haspin. In summary, we conclude that CPC-mediated localisation of Aurora B activity to pericentromeres rather than bulk Aurora B activity is crucial for recruitment of the spindle checkpoint kinase MPS1 to kinetochores.

## Structure of the CPC targeting module with T3-phosphorylated histone H3 nucleosomes

Progress in understanding chromatin binding by the CPC at a molecular level has been limited by the ability to produce large quantities of H3pT3 modified nucleosomes mimicking those at the pericentromere. As an alternative to small-scale chemical ligation-based methods for production of H3pT3 nucleosomes, the kinase domain of Haspin was recombinantly expressed, purified, and used to stoichiometrically phosphorylate recombinant histone H3 (Appendix Fig. S3A,B). The resulting H3pT3 was directly used to assemble large amounts of H3pT3 nucleosomes for cryo-EM and biochemical studies (Appendix Fig. S3C). Two variants of the CPC targeting module with full-length borealin containing the C-terminal dimerization domain (Appendix Fig. S3D, CPC80 Borealin$_{(FL)}$) or a truncated version lacking that domain but encompassing the minimal region required for centromeric targeting (Appendix Fig. S3D, CPC80 Borealin$_{(1-76)}$ and Appendix Fig. S4) were then tested for binding to nucleosomes containing histone H3 or T3-phosphorylated histone H3 using electrophoretic mobility shift assays (EMSAs). Monomeric CPC80 Bor$_{(1-76)}$ bound poorly to non-phosphorylated nucleosomes even at high concentrations equivalent to 16-fold molar excess (Fig. EV2A). In agreement with the idea that the CPC targets to a specific subset of pericentromeric nucleosomes in cells marked by haspin activity, good binding was observed from >fourfold molar excess when T3-phosphorylated nucleosomes were used (Fig. EV2A). Dimeric CPC80 Bor$_{(FL)}$ resulted in a supershift of the nucleosomes into large complexes or chains that failed to enter the gel, and again this effect was markedly stronger when T3-phosphorylated nucleosomes were used (Fig. EV2A). Four different binding modes can be envisaged for CPC interaction with T3-phosphorylated nucleosomes (Fig. EV2B). Based on the biochemical data we favour the idea that the CPC binds to a single face on each nucleosome, and dimeric CPC bridges between nucleosomes rather than acting as a clamp capturing a single nucleosome. To understand these potential modes of binding we therefore set out to determine the structures of dimeric and monomeric CPC complexes with

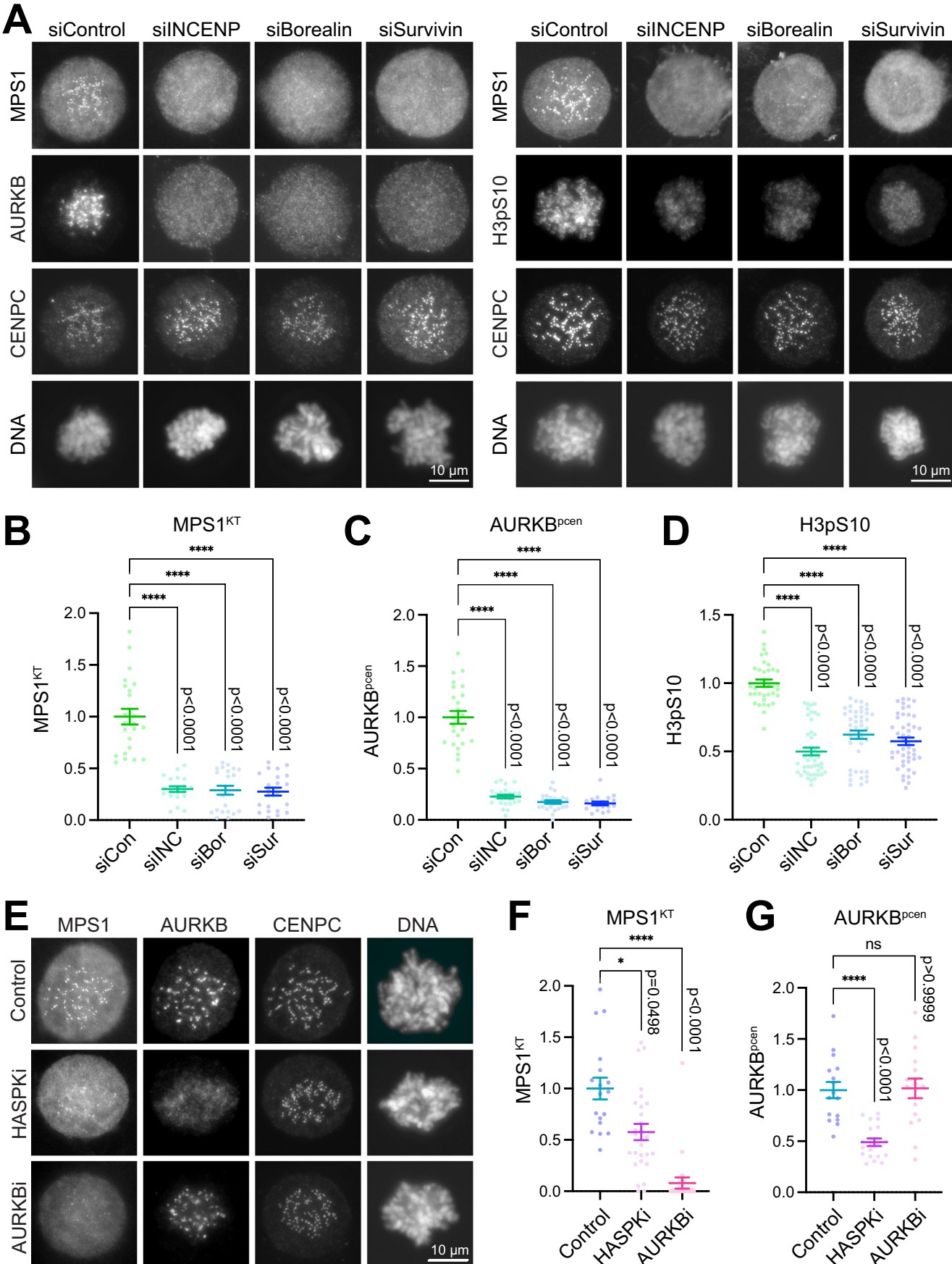

**Figure 2. Kinetochore recruitment of MPS1 requires pericentromeric localisation of the CPC.**

(A) Dependence of MPS1 kinetochore localisation (MPS1$^{KT}$), Aurora B pericentromere (AURKB$^{pcen}$), and histone H3pS10 levels on INCENP (ICP), borealin (Bor), or survivin (Sur) was tested using siRNA depletion in prometaphase-arrested HeLa MPS1-GFP cells. Representative images with scale bars of 10 μm are shown. (B–D) Scatter plots with mean ± SEM of MPS1$^{KT}$, AURKB$^{pcen}$ and H3pS10 with non-parametric Kuskal–Wallis tests with Dunn's multiple comparison test for significance, $P < 0.0001$ (****) for all comparisons indicated on the graphs. Samples sizes are (B) $n > 19$ cells, (C) $n > 18$ cells, (D) $n > 43$ cells in all cases from at least three independent experiments. (E) Dependence of MPS1$^{KT}$ and pericentromeric Aurora B (AURKB$^{pcen}$) on Haspin or Aurora B kinase activities were tested using 5 min nocodazole treatment of MG132-arrested cells to reactivate the spindle assembly checkpoint in the presence of specific inhibitors, HASPKi and AURKBi, respectively, or DMSO as a control. Representative images with a scale bar of 10 μm are shown. (F, G) Scatter plots with mean ± SEM for MPS1$^{KT}$ and AURKB$^{pcen}$ with non-parametric Kuskal–Wallis tests with Dunn's multiple comparison test for significance, $P < 0.0001$ (****), $P = 0.0498$ (*), $P > 0.9999$ (ns) for the comparisons indicated in the graphs. Sample sizes were (F) $n > 18$ cells and (G) $n > 17$ cells from at least three independent experiments.

nucleosomes. Initial attempts at determining a CPC-H3pT3 nucleosome structure by cryo-EM used a CPC construct consisting of Survivin$_{(FL)}$, INCENP$_{(1-80)}$, and full-length Borealin$_{(FL)}$. These particles appeared heterogenous and had a propensity to aggregate, issues which were alleviated by BS3-crosslinking and addition of a detergent 0.3% β−octylglucoside to the final sample (Fig. EV3A), thus enabling data collection and assignment of 2D classes (Fig. EV3B). Within the resulting map we could clearly identify regions corresponding to every component of the nucleosome, as well as an additional density formed by a single bound CPC molecule (Fig. EV3C–F). Although only a single nucleosome is captured in the structure, this mode of binding is more consistent with bridging rather than a clamp model (Fig. EV2B). An initially striking and unexpected feature present in the map was the partial unwrapping of the DNA from around the histone octamer core. Since the CPC did not appear to make any contacts with the DNA in the map, we suspected that this might be caused by the BS3 crosslinking agent acting on the nucleosomes. This was tested by collecting small cryo-EM datasets of DNA wrapped nucleosomes before and after BS3-crosslinking (Fig. EV3G, -BS3 crosslinking). DNA unwrapping was observed in the BS3 crosslinked sample, supporting the view that DNA unwrapping was caused by the BS3 crosslinker and not due to the CPC. In the combined BS3-crosslinked CPC-H3pT3 nucleosome sample, the local resolution of map regions corresponding to CPC was relatively low in comparison to the histone octamer, at 8–12 Å versus 4–6 Å, respectively (Fig. EV3D–F). This suggested that the CPC was undergoing movement relative to the nucleosome. The density attributed to CPC corresponded closely with the previously crystallised minimal constructs of the CPC localisation module (Jeyaprakash et al, 2007; Serena et al, 2020), and appeared to make contact with the surface of the histone octamer. Densities that might belong to INCENP RRKKRR motif or the extended C-terminal region and dimerization domains of borealin were not observed, suggesting that they do not adopt a fixed conformation under these conditions.

Together, these results indicated that crosslinking with BS3 was not suitable for imaging the CPC-H3pT3 nucleosome complex with the DNA wrapped in its native state. This meant we required a revised strategy to reduce sample heterogeneity on cryo-EM grids, which we suspected was in part caused by the borealin loop and C-terminal dimerization domains. Because these regions of borealin were not observable in the initial cryo-EM maps, we decided to proceed with a different construct for the CPC targeting module containing a truncated version of borealin containing only residues 1–76, which forms the minimal domain required for

centromeric localisation (Appendix Fig. S4), and to omit the BS3-crosslinking step.

## Borealin N-terminus binds the nucleosome acidic patch

A modified CPC construct (INCENP$_{(1-80)}$, Survivin$_{FL}$, and Borealin$_{(1-76)}$) was used to prepare CPC-H3pT3 nucleosome complexes on cryo-EM grids for data collection and processing (summarised in Appendix Fig. S5 and Table EV1). Based on this data, a combined model for the minimal CPC localisation module in complex with a H3pT3 nucleosome was assembled and is show in a nucleosome top and side view (Fig. 3A). As explained in detail in the following sections the region corresponding to the CPC targeting module was of slightly lower resolution (Fig. 3B), however the previously determined CPC crystal structure fitted well within the map (Fig. 3C) and overall resolution was 2.4 Å (Fig. 3D,E). Initial processing of the collected dataset produced a map at 2.3 Å global resolution with minimal processing (Appendix Fig. S5). The H3pT3 nucleosome can be clearly identified in the map, as well as an additional density proximal to the acidic patch formed by the interface between histones H2A and H2B on the surface of the histone octamer. Density corresponding to the CPC construct was present during the intermediate processing steps but was masked out during the final non-uniform refinement due to poor signal strength in comparison to the rest of the map. By fitting CPC in the intermediate maps, we could unambiguously identify this additional density as the N-terminus of borealin (Fig. EV4A–E). As previously reported using a crosslinking approach and confirmed here in the molecular structure, the borealin N-terminus contains several arginine and lysine residues that enable it to bind the H2A–H2B acidic patch on the nucleosome surface via salt bridges (Fig. 4A–C) (Abad et al, 2019). Based on sidechain signal strength, the residues most important for the interaction between borealin and the H3pT3 nucleosome appeared to be R4, R9, and K12 (highlighted in Fig. 4B,C, panels 1–3). R4 interacts with Q47 and E113 on H2B, as well as E56 on H2A, thus stabilising the interaction at the very end of the borealin N-terminus, whereas other minor interactions appear to be formed between K5 and H2A E64, as well as S7 and H2B H109 (Fig. 4C, panels 1–2). The main interaction hotspot between borealin and the nucleosome is formed by R9 and K12, which are located at the very beginning of the borealin α-helix taking part in the CPC helical bundle (Fig. 4C, panel 3). R9 acts as an anchor forming interactions with three different aspartate and glutamate residues on H2A (E92, D90, E61), while K12 further stabilises this site by an additional interaction with E61, as well as H2A E64 already bound by K5. Furthermore,

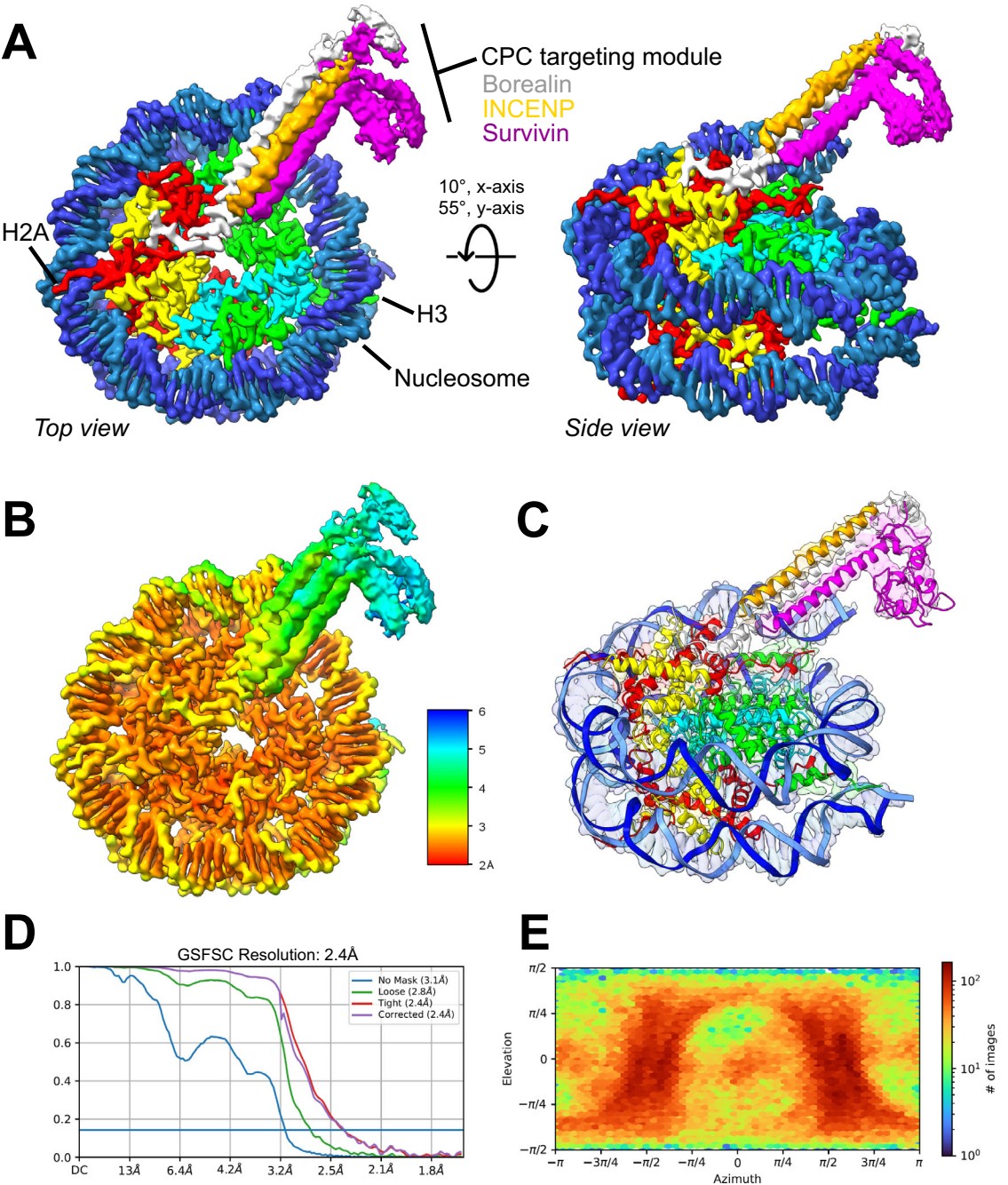

**Figure 3. Structure of the CPC bound to nucleosomes containing T3-phosphorylated histone H3.**

(A) Cryo-EM density of the CPC targeting module in complex with an H3pT3 nucleosome with Histone H2A (red), H2B (yellow), H3 (green), H4 (cyan), DNA (blue), and the CPC subunits borealin (grey/white) survivin (magenta), INCENP (gold/orange). A top view looking at one face of the nucleosome and a slightly rotated side view are shown. The positions where the H2A and H3 tails emerge from the nucleosome are indicated. EMBD ID - EMD-19685, refinement after 3D classification, Table EV1B. See also PDB ID 8RUP, EMDB ID—EMD-19513 and, Table EV1D (final). (B) A local resolution map for the CPC-H3pT3 complex. (C) A sideview of the fitted model matching the side view in (A). (D) Gold standard Fourier shell correlation (GSFSC) resolution, blue horizontal line indicates an FSC value of 0.143, and (E) particle orientation distribution within the final map.

K12 is also capable of binding the backbone carbonyl group of borealin residue G6, thus potentially stabilising the bend within the flexible borealin N-terminal loop just before its first helix.

Guided by this data, we tested the importance of both R9 and K12 by mutation to alanine and performing EMSA studies with these mutant CPC complexes and the H3pT3 nucleosome (Fig. 4D). For all mutants tested, the distance of band shifts present at each CPC concentration was reduced in comparison to the wild-type borealin construct (Fig. 4D, borealin$_{(1-76)}$ R9A and K12A). This effect was more pronounced in the double-mutant (Fig. 4D,

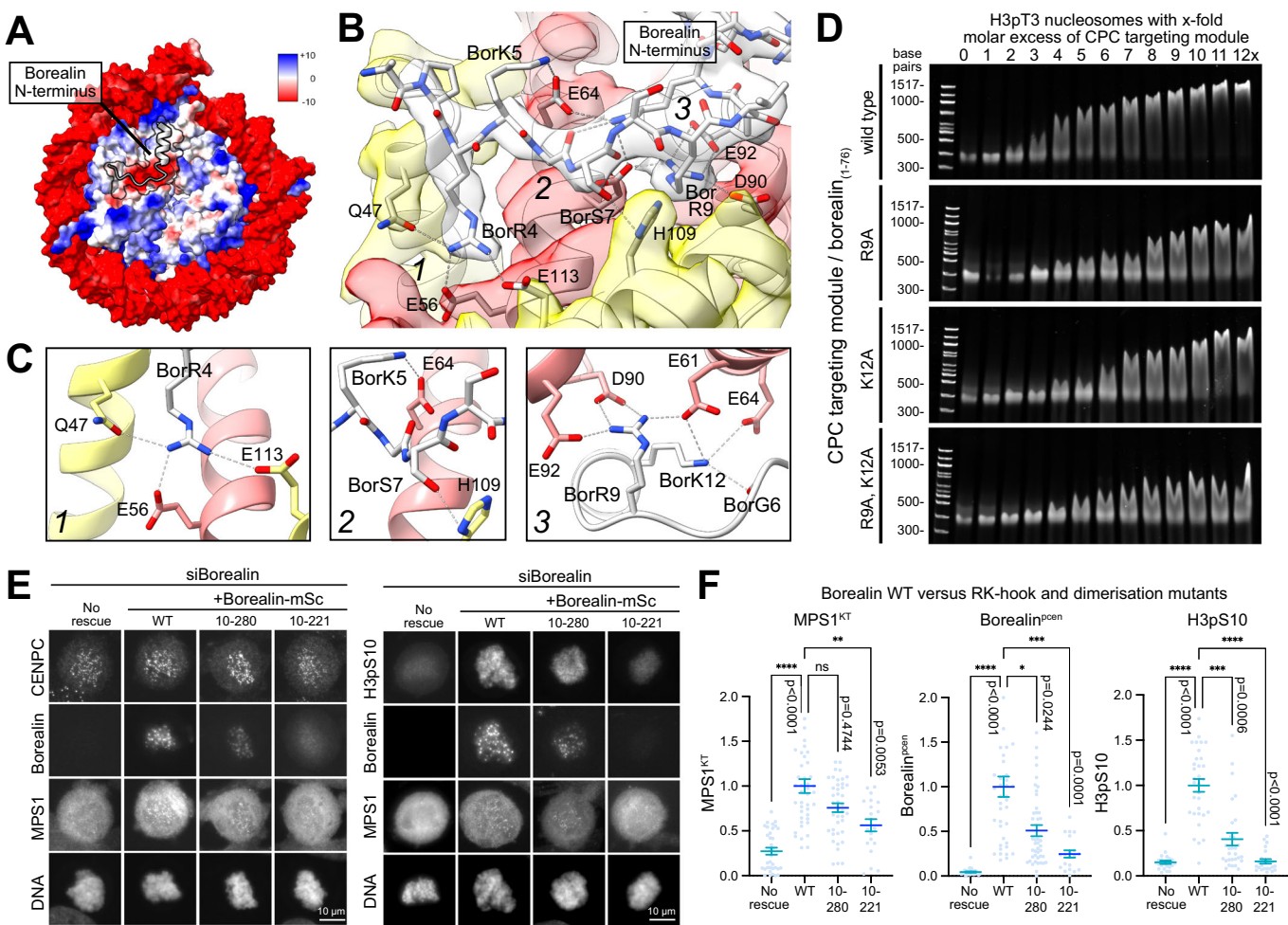

**Figure 4. Borealin-nucleosome acidic patch interaction is necessary for MPS1 recruitment.**

(A) A Coulombic electrostatic potential map of the CPC targeting module in complex with an H3pT3 nucleosome. PDB ID 8RUQ, EMDB ID—EMD-19514, Table EV1A (all particles). (B) The borealin N-terminus (white) interacts with the acidic patch (black outline indicates idealised surface) formed by H2A (red) and H2B (yellow). Numbers indicate three areas of interest. (C) Details of the interactions between the borealin N-terminus (grey/white) and different regions 1–3 of the nucleosome acidic patch. (D) EMSAs between 150 nM H3pT3 phosphorylated nucleosomes and CPC80-Bor$_{(1-76)}$ targeting module complexes containing borealin$_{1-76}$ wild-type, R9A, K12A or R9A-K12A mutations at the specified molar excess. (E) Dependence of MPS1 kinetochore localisation (MPS1$^{KT}$), pericentromeric enrichment of borealin-mScarlet (borealin$^{pcen}$) and H3pS10 on chromatin were tested in HeLa Flp-In T-REx MPS1-GFP cells using siRNA depletion and rescue assays for wild-type borealin, an N-terminal deletion lacking the first nine amino acids (10–280) and an N-terminal deletion lacking the first nine amino acids and the C-terminal dimerization domain (10–221). Representative images with 10 μm scale bars are shown. (F) scatter plots with mean ± SEM for MPS1$^{KT}$, borealin$^{pcen}$ and H3pS10 and a Kruskal–Wallis test and Dunn's test for multiple comparisons for significance for comparisons indicated in the graphs. For MPS1$^{KT}$, No rescue vs WT $P < 0.0001$ (****), WT vs 10–281 $P = 0.4744$ (ns) and WT vs 10–221 $P = 0.0053$ (**). For Borealin$^{pcen}$, No rescue vs WT $P < 0.0001$ (****), WT vs 10–281 $P = 0.0244$ (*) and WT vs 10–221 $P = 0.0001$ (***). For H3pS10, No rescue vs WT $P < 0.0001$ (****), WT vs 10–281 $P = 0.0006$ (***) and WT vs 10–221 $P < 0.0001$ (***). Sample sizes (F) $n > 20$ cells MPS1, $n > 20$ cells borealin, $n > 23$ cells H3pS10 from at least three independent experiments.

borealin$_{(1-76)}$ R9A–K12A), thus showing that both R9 and K12 have a role in establishing the interaction between CPC and H3pT3 nucleosome. In cells, truncation of this Arginine-Lysine (RK)-anchor region of borealin resulted in the reduction of both MPS1 at kinetochores and H3pS10 on chromatin in prometaphase arrested cells when compared to cells expressing wild-type borealin (Fig. 4E,F, MPS1$^{KT}$ and H3pS10, WT vs 10–280). In agreement with those effects, recruitment of the RK-anchor defective form of borealin to the pericentromeres was reduced compared to the wild-type protein but not abolished (Fig. 4E,F, Borealin$^{pcen}$, WT vs 10–280). Borealin recruitment to pericentromeres was further reduced by combined deletion of the RK-anchor region and

dimerization domain (Fig. 4E,F, Borealin$^{pcen}$, WT vs 10–221) in line with the view that dimerization increases the avidity for chromatin through interactions independent of the borealin RK anchor (Abad et al, 2019; Bekier et al, 2015; Bourhis et al, 2009; Trivedi et al, 2019; Trivedi and Stukenberg, 2020). Thus, while the RK anchor region is crucial for full pericentromere targeting of the CPC, recruitment of MPS1 and efficient phosphorylation of histone H3 at S10 at non-centromeric chromatin, in agreement with previous work it is unlikely to be the sole determinant of chromatin binding. Since, the nucleosome acidic patch is not a unique feature of pericentromeres the role of additional interactions between survivin and the H3pT3 mark on pericentromeric nucleosomes was explored next.

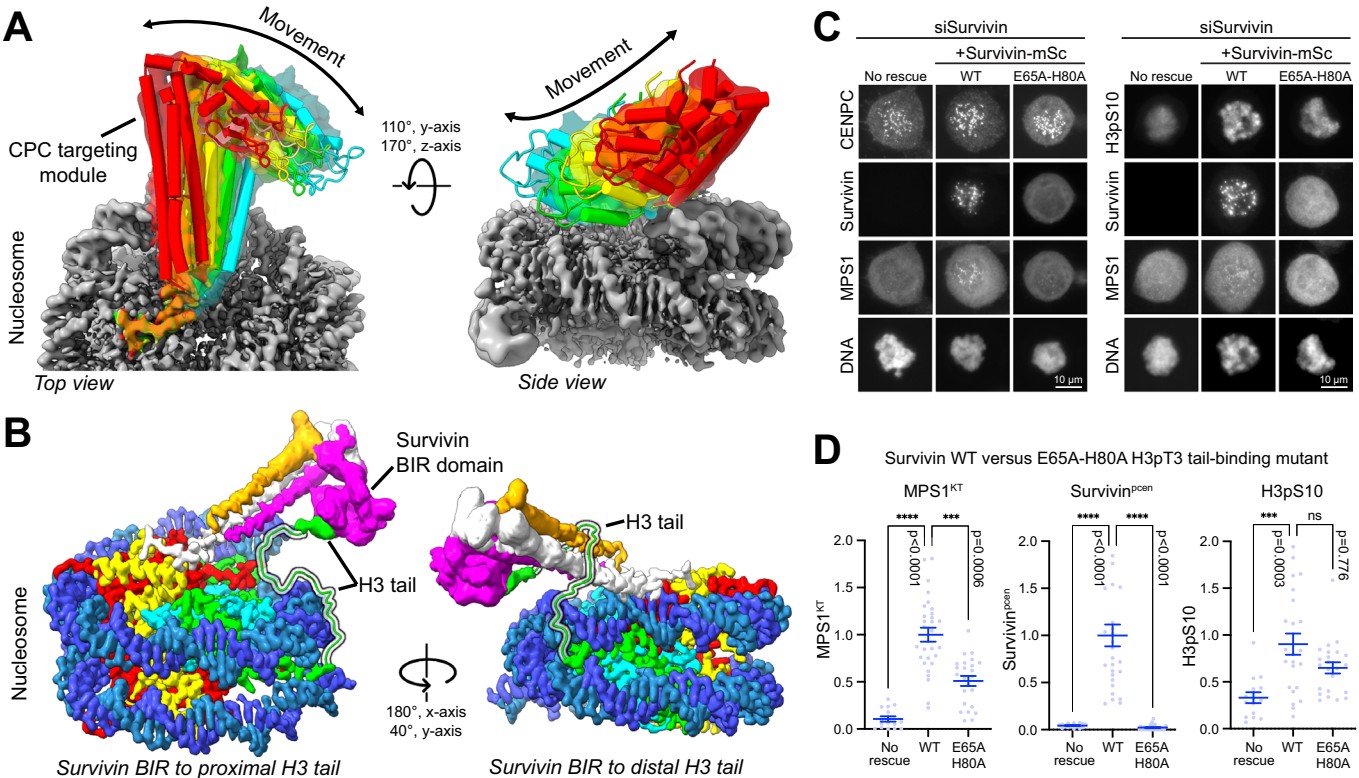

Figure 5. CPC-nucleosome histone H3 tail interactions are important for MPS1 recruitment.

(A) Selected classes from 3D classification are depicted (Appendix Fig. S5). The CPC targeting module pivots around the borealin N-terminus bound to H3pT3 nucleosome surface. Red—class 4 (10.4% of particles); yellow—class 6 (10.0%), green—class 2 (10.5%); cyan—class 5 (10.0%). (B) H3pT3 nucleosome with Histone H2A (red), H2B (yellow), H3 (green), H4 (cyan), DNA (blue), and the CPC subunits borealin (grey/white), survivin (magenta), INCENP (gold/orange). PDB ID 8RUP, EMDB ID—EMD-19513 and, Table EV1D (final). Potential trajectories of H3 tail bound by CPC are depicted. CPC can be bound by H3 tails (highlighted) emerging either from the proximal nucleosome face bound by CPC (left), or from the opposite or distal face (right). The latter binding mode appears to be sterically hindered due to length constraints imposed by the histone H3 tail. (C) Dependence of MPS1 kinetochore localisation (MPS1$^{KT}$), pericentromeric enrichment of survivin-mScarlet (survivin$^{pcen}$) and H3pS10 on chromatin were tested in HeLa Flp-In T-REx MPS1-GFP cells using siRNA depletion and rescue assays for wild-type survivin and histone H3pT3 binding E65-H80A point mutant. Representative images with 10 μm scale bars are shown. (D) scatter plots with mean ± SEM for MPS1$^{KT}$, survivin$^{pcen}$ and H3pS10 with a Kruskal–Wallis test and Dunn's test for multiple comparisons for significance as indicated on the graphs. For MPS1$^{KT}$, No rescue vs WT $P < 0.0001$ (****) and WT vs E65A H80A $P = 0.0006$ (***). For Survivin$^{pcen}$, No rescue vs WT $P < 0.0001$ (****) and WT vs E65A H80A $P < 0.0001$ (****). For H3pS10, No rescue vs WT $P = 0.0003$ (***) and WT vs E65A H80A $P = 0.776$ (ns). Samples sizes were $n > 15$ cells for MPS1$^{KT}$, Survivin$^{pcen}$ and H3pS10 from at least three independent experiments.

## CPC undergoes rotational movement around the borealin pivot in the nucleosome-bound state

Although the borealin N-terminal RK-anchor was observable within the initial map (Appendix Fig. S5), as already mentioned, the strength of signal corresponding to CPC was relatively weak. We hypothesised that the CPC, although a rigid structure itself, can move relative to the nucleosome in its bound state. To explore this idea, localised 3D-classification focusing on the CPC rather than the H3pT3 nucleosome density was performed to give a medium resolution reconstruction of 6–7 Å. This generated an equal distribution of classes, with CPC either being absent or found in a variety of slightly different orientations (Fig. 5A). By aligning classes exhibiting well-defined CPC density, it was apparent that the CPC undergoes a swivel motion relative to the nucleosome surface (Fig. 5A). The borealin N-terminus acts as a pivot or anchor point that remains unchanged across the classes observed, consistent with the density proximal to the nucleosome acidic patch seen in our initial maps (Figs. 4A and EV4A–C). The

presence of this mode of movement was also supported by 3D variability analysis.

## The CPC binds to the phosphorylated H3 tail emerging from the proximal face of the nucleosome

To understand how the histone H3 tail engages with the CPC, the class exhibiting the highest local resolution for CPC was selected for further processing. This class corresponded to a state close to the middle point of the CPC swivel motion. Going from the H3pT3 nucleosome-proximal to the distal end of the CPC, the local resolution gradually worsened, with the survivin BIR domain being the most poorly defined region of the map (Fig. EV5). This was to be expected, since the distal end of CPC would exhibit the most degrees of freedom due to it being furthest from the borealin N-terminus anchored to the nucleosome surface at the H2A–H2B acidic patch. To further resolve the CPC, particle subtraction and localised refinement focusing only on the distal end of CPC was performed (Fig. EV5A). Despite a reduction in the reported local

resolution values, this procedure gave rise to a clearer, more connected map density allowing us to confidently identify and place the secondary structure elements of the BIR domain. Furthermore, the new map also contained an additional density corresponding to the T3-phosphorylated H3 tail (Fig. EV5B–F), in a position that matches the previously reported structures obtained by X-ray crystallography (Serena et al, 2020).

Despite further processing efforts aiming to resolve the rest of the H3 tail, it remained absent in the final map, therefore suggesting that it remains flexible in its CPC-bound state. Due to the two-fold symmetry of the nucleosome, there are two H3 tails in the vicinity of the survivin BIR domain, extending out from either the CPC-bound proximal or unbound distal face of the H3pT3 nucleosome. To determine which was more likely to bind the CPC, as well as approximate their possible orientation within the CPC-bound state, both these distal and proximal alternatives were modelled in the final structure (Fig. 5B, distal and proximal). The resulting models showed that the H3 tail extending from the CPC-bound proximal face can easily reach its binding site within the BIR domain, thus allowing it to remain flexible upon interaction (Fig. 5B, proximal). In contrast, the H3 tail emerging from the opposite or distal face of the nucleosome was not able to reach its binding site via a path going under the BIR domain. Length constraints dictate a path going over the CPC helical bundle (Fig. 5B, distal), which would also reduce its flexibility considerably, meaning it should have been able to resolve the density in the cryo-EM map, which was not the case. Based on these models we therefore propose that CPC interacts with the T3-phosphorylated H3 tail extending from the same face bound by borealin N-terminus. This allows both CPC and the H3 tail to remain flexible in their bound states, which might be important for maintaining nucleosome surface access for other nucleosome- or CPC-binding proteins, such as Sgo1 or MKLP2.

Finally, the role of the interaction between the survivin BIR domain and the histone H3 tail was tested using structure guided mutants characterised previously (Jeyaprakash et al, 2011; Serena et al, 2020). Accordingly, E65 and H80 in the survivin BIR domain were mutated in combination to disrupt binding to histone H3 R2 and the peptide backbone adjacent to A1, and the effects on MPS1 recruitment and pericentromeric enrichment of Aurora B were tested. When compared to wild-type survivin, the E65-H80 mutant supported significantly less recruitment of MPS1 to kinetochores and CPC to the pericentromeres (Fig. 5C,D, Survivin WT vs E65-H80, MPS1$^{KT}$, Survivin$^{pcen}$). In contrast, H3pS10 a marker for Aurora B activity at non-centromeric chromatin was not significantly changed in cells expressing the survivin E65-H80 mutant (Fig. 5C,D, Survivin WT vs E65-H80, H3pS10). Together, these findings support the idea that the correct spatial restriction of the CPC to pericentromeres, and thus localised Aurora B activity, are critical for spindle checkpoint activation but are less crucial for global Aurora B activity towards non-centromeric chromatin.

## Discussion

Taken together our data show that the pericentromeric localisation of the CPC is mediated by the combined interactions of borealin with the H2A-H2B acidic patch on the nucleosome surface to create a fixed pivot point, and the interaction of the histone H3 tail

with the survivin BIR domain creating a flexible tether. Both these interactions are necessary to correctly localise Aurora B to pericentromeres and to trigger recruitment of the spindle checkpoint kinase MPS1 in prometaphase cells. The microtubule driven physical separation of kinetochores away from this CPC-dependent pericentromeric pool of Aurora B at bioriented metaphase chromosomes helps explain why MPS1 is not recruited in that state. This arrangement creates a simple physical mechanism to discriminate bioriented chromosomes, at which kinetochores and the CPC are spatially separated, from error states triggering the spindle assembly checkpoint, at which they overlap (Lampson and Cheeseman, 2011; Liu et al, 2009). Only in the latter "collapsed" state is sustained recruitment of MPS1 to kinetochores observed.

Our cryo-EM structures of the CPC localisation module in complex with an H3pT3-nucleosome show that the N-terminus of borealin is the only region of the CPC forming a conformationally stable interaction with nucleosomes (Figs. 4A and EV4A–C). This is in close agreement with previous work using other approaches (Abad et al, 2019) and a recent preprint describing a similar cryo-EM structure (preprint: Gireesh et al, 2025). A positively charged loop region located at the N-terminus of borealin contains several key residues, such as R4, R9, and K12, which participate in a network of salt bridges with residues found on the acidic patch of nucleosomes found at the interface between H2A and H2B (Fig. 4B,C). The acidic patch of nucleosomes is crucial for their recognition by most nucleosome-binding proteins (McGinty and Tan, 2021), and our results show that the CPC is no exception. This interaction is independent of any cell cycle-dependent post-translational modifications, thus indicating that borealin ensures an intrinsic propensity of the CPC to localise to nucleosomes. The reduction in H3pS10 in cells expressing mutants of borealin lacking the RK anchor (Fig. 4D) suggest that this interaction is important to drive Aurora B phosphorylation at non-centromeric chromatin.

Despite the importance of the borealin N-terminal RK anchor and C-terminal dimerization domain, both our EMSA results and other previously published reports have shown that other regions of the CPC localisation module, such as the BIR domain of survivin binding the H3 tail, as well as the positive RRKKRR motif of INCENP, are important for efficient nucleosome binding and CPC localisation during different stages of mitosis (Jeyaprakash et al, 2007; Serena et al, 2020). In contrast to the RK-anchor region of borealin, our analysis shows that the BIR domain and the H3 tail can undergo movement in the bound state relative to the core nucleosome (Fig. 5A). Similarly, we could not observe the C-terminal loop region of INCENP containing the positive RRKKRR motif. This strongly suggests that it does not have a single well-defined interaction site on the surface of nucleosomes. Since it was previously shown to bind DNA in vitro and contribute to CPC targeting in vivo (Serena et al, 2020), it is likely that the positive motif on INCENP non-specifically interacts with negatively charged DNA strands wrapped around the core of the nucleosome in a variety of different orientations. Therefore, it appears that interactions between the nucleosome and either survivin or INCENP do not contribute to stabilisation of the CPC in a specific conformational state, instead only acting to tether the CPC to the surface of the nucleosome while allowing it to remain flexible.

As indicated by our data, the CPC exhibits a high degree of flexibility relative to the surface of the nucleosome, with the borealin N-terminus acting as the pivot point. Since the functional kinase module of CPC is attached to its localisation module via a long, flexible linker formed by INCENP, there is no apparent reason for why CPC should be attached to the surface of nucleosomes in a single, well-defined orientation. Instead, its flexibility might serve to permit unrestricted nucleosome access to other nucleosome-binding proteins required for mitosis. These would include histone tail-modifying enzymes that, in addition to regulating overall mitotic progression, would also directly influence CPC localisation. Furthermore, the binding modes of Sgo1 and MKLP2, two key proteins regulating the CPC after its interaction with the phosphorylated H3 tail, are only partly understood (Abad et al, 2022; Serena et al, 2020). It is possible that the flexibility of the CPC relative to the nucleosome is required to contact these proteins, thus enabling changes in its localisation to occur more efficiently.

As nucleosomes exhibit two-fold symmetry, we expected to observe 3D classes where a single T3-phosphorylated nucleosome would be bound or clamped by two CPC molecules, but this was not the case. During 2D-classification, we did note a few classes where additional density was seen on both sides of the nucleosome. This might be attributed to the classification software focusing mainly on the regions of strongest signal present, corresponding to H3pT3 nucleosome, without fully aligning the relatively weak CPC signal. However, attempts to impose C2 symmetry during the refinement process resulted in diminished CPC density, suggesting that only a single bound molecule of CPC was present in our dataset, despite the CPC being added to H3pT3 nucleosome in excess. Despite this conclusion, the final maps indicate that binding of a single CPC molecule should not sterically hinder the binding of another one on the opposite surface of the nucleosome. Thus, we cannot rule out that a clamp binding mode facilitated by borealin dimerization could exist in vivo. Alternatively, as our biochemical data suggest, the CPC may bridge between nucleosomes to create chains in a manner dependent on borealin dimerization and interaction of survivin with the H3pT3 modification (Fig. EV2A,B). Therefore, while our study provides important information about the interaction of the CPC with individual nucleosomes, further

work will be important to address how the CPC interacts with nucleosomes in the context of chromatin in vivo and the potential for different clamping and bridging modes of interaction.

In summary, our results shed light on one of the several possible CPC binding modes on the surface of nucleosomes, which corresponds to the pool of the CPC bound mostly to pericentromeres during prometaphase and metaphase (Yamagishi et al, 2010). This leaves a series of important unanswered questions relating to chromatin structure and the role of other CPC-associated proteins. To further our understanding of CPC localisation during these stages of mitosis, the primary target for subsequent structural and biochemical analyses would be the Sgo1-CPC-H3pT3 nucleosome complex. Sgo1 has also been shown to be important for CPC localisation in cells delayed in mitosis for long periods, interacting with both core CPC components and H2ApT120 (Broad et al, 2020; Hadders et al, 2020; Kawashima et al, 2007). Other studies have suggested that Sgo1 might compete for the H3pT3 binding site in survivin, while also potentially interacting with the CPC helical bundle (Abad et al, 2022). Interestingly, we see that Sgo1 localises in two peaks adjacent to the centromere and kinetochores, flanking a pericentric domain containing Aurora B and the CPC (Appendix Fig. S1A). This raises the possibility that Sgo1 may help restrict CPC localisation by acting as a boundary element, rather than acting as a primary binding determinant. To fully understand CPC function and regulation, it will therefore be necessary to determine further structures and analyse how binding partners such as Sgo1 affect CPC orientation relative to the nucleosome. We have not explored the impact of CPC binding on chromatin structure since our study has used single nucleosomes, and further studies are therefore required to resolve these important questions. Similarly, a structure of the CPC and MKLP2 complex will help explain its rapid removal from chromosomes at the onset of anaphase. Answers to these questions will allow us to construct a detailed overview of CPC localisation throughout mitosis, thus leading to a comprehensive understanding of the modes of function of one of the key players required for faithful chromosome segregation during cell division.

# Methods

**Reagents and tools table**

| Reagent/resource | Reference or source | Identifier or catalog number |
|---|---|---|
| **Experimental models** | | |
| HCT116 (*H. sapiens*) | ATCC | Cat: #CCL-247 |
| HeLa (*H. sapiens*) | ATCC | Cat: #CRL-2.2 |
| HeLa Flp-In T-REx (*H. sapiens*) | Hewitt et al, 2010 | |
| HCT116 NDC80-HaloTag | This study | |
| HeLa MPS1-GFP | Alfonso-Perez et al, 2019 | |
| HeLa Flp-In T-REx MPS1-GFP | This study | |
| HeLa Flp-In T-REx MPS1-GFP Borealin WT-mScarlet | This study | |
| HeLa Flp-In T-REx MPS1-GFP Borealin 10–281-mScarlet | This study | |
| HeLa Flp-In T-REx MPS1-GFP Borealin 10–220-mScarlet | This study | |
| HeLa Flp-In T-REx MPS1-GFP Survivin WT-mScarlet | This study | |
| HeLa Flp-In T-REx MPS1-GFP Survivin E65A H80A-mScarlet | This study | |
| XL-1 Blue (*E. coli*) | Agilent | Cat: #200249 |
| BL21(DE3) pLysS (*E. coli*) | Invitrogen | Cat: #C606010 |

| Reagent/resource | Reference or source | Identifier or catalog number |
|---|---|---|
| BL21-CodonPlus (DE3)-RIL (*E. coli*) | Agilent | Cat: #230245 |
| **Recombinant DNA** | | |
| pX459 | Addgene | Cat: #62988 |
| pSpCAS9(BB) | Addgene | Cat: #48139 |
| pcDNA5/FRT/TO | Thermo Fisher | Cat: #V652020 |
| pSC-A | Agilent | Cat: #240207 |
| pUC19 Widom 601 ×16 | This study | |
| pET3a histone H2A | Luger et al, 1997a, b | |
| pET3a histone H2B | Luger et al, 1997a, b | |
| pET3a histone H3 | Luger et al, 1997a, b | |
| pET3a histone H4 | Luger et al, 1997a, b | |
| pNIC28-Bsa4 vector containing human Haspin residues 465–798 | Addgene | Cat: #38915 |
| pETDuet-1 Survivin 1–142/His6-borealin 1–280 | This study | |
| pETDuet-1 Survivin 1–142/His6-borealin 1–76 | This study | |
| pETDuet-1 Survivin 1–142/His6-borealin 1–76 R9A | This study | |
| pETDuet-1 Survivin 1–142/His6-borealin 1–76 R12A | This study | |
| pETDuet-1 Survivin 1–142/His6-borealin 1–76 R9A R12A | This study | |
| pFAT2 His$_6$-GST INCENP 1–80 | This study | |
| pcDNA5/FRT/TO Survivin WT-mScarlet | This study | |
| pcDNA5/FRT/TO Survivin E65A H80A-mScarlet | This study | |
| pcDNA5/FRT/TO Borealin WT-mScarlet | This study | |
| pcDNA5/FRT/TO Borealin 10–281 -mScarlet | This study | |
| pcDNA5/FRT/TO Borealin 10-220 -mScarlet | This study | |
| **Antibodies** | | |
| Aurora B (STED 1:100) | BD Biosciences | RRID:AB_398396<br>Cat: #611083 |
| Aurora B (STED 1:100) | Abcam | RRID:AB_302923<br>Cat: #ab-2254 |
| Borealin (STED 1:100) | Santa Cruz | RRID:AB_11151778<br>Cat: #sc-376636 |
| Survivin (STED 1:250) | Abcam | RRID:AB_3662145<br>Cat: #ab-134170 |
| α-Tubulin (STED 1:250) | ABCD | RRID:AB_3106979<br>Cat: #ABCD_AA345 |
| CENPA (STED 1:250) | GeneTex | RRID:AB_369391<br>Cat: #GTX13939 |
| Sgo1 (STED 1:100) | Abcam | RRID:AB_945427<br>Cat: #ab-58023 |
| CF680R anti-rabbit (STED 1:250) | Biotium | RRID:AB_10852822<br>Cat: #20195-1 |
| CF680R anti-mouse (STED 1:250) | Biotium | RRID:AB_10854861<br>Cat: #20194-1 |
| Alexa Fluor 594 donkey anti-rabbit | Invitrogen | RRID:AB_2762827<br>Cat: #A-32754 |
| Alexa Fluor 594 donkey anti-mouse (STED 1:100-1:250) | Invitrogen | RRID:AB_2762826<br>Cat: #A-32744 |
| INCENP (IF/WB 1:2000) | Santa Cruz | RRID:AB_11149761<br>Cat: #sc-376514 |
| Borealin (IF/WB 1:2000) | Santa Cruz | RRID:AB_11151778<br>Cat: #sc-376635 |
| Survivin (IF/WB 1:5000) | Abcam | RRID:AB_1524459<br>Cat: #ab-76424 |
| CENPC (IF/WB 1:2000) | MBL International | RRID:AB_10693556<br>Cat: #PD030 |
| Aurora B (IF/WB 1:2000) | Cell Signalling Technology | RRID:AB_10695307<br>Cat: #3094S |
| H3pS10 (IF/WB 1:40000) | Cell Signalling Technology | RRID:AB_331748<br>Cat: #9706S |
| Alexa Fluor 555 donkey anti-rabbit (IF 1:1000) | Thermo Fisher | RRID:AB_162543<br>Cat: #A-31572 |
| Alexa Fluor 555 donkey anti-mouse (IF 1:1000) | Thermo Fisher | RRID:AB_2536180<br>Cat: #A31570 |
| Alexa Fluor 555 donkey anti-rabbit (IF 1:1000) | Thermo Fisher | RRID:AB_2536183<br>Cat: #A31573 |

| Reagent/resource | Reference or source | Identifier or catalog number |
|---|---|---|
| Alexa Fluor 647 donkey anti-mouse (IF 1:1000) | Thermo Fisher | RRID:AB_162542<br>Cat: #A31571 |
| Alexa Fluor 647 goat anti-guinea pig (IF 1:1000) | Thermo Fisher | RRID:AB_141882<br>Cat: #A-21450 |
| Alexa Fluor 555 goat anti-guinea pig (IF 1:1000) | Thermo Fisher | RRID:AB_2535856<br>Cat: #A-21435 |
| Histone H3 (WB 1:1000) | Cell Signalling, | RRID:AB_3289584<br>Cat: #60932 |
| T3 phosphorylated histone H3 (WB 1:1000) | Millipore | RRID:AB_310604<br>Cat: #07-424 |
| Peroxidase donkey anti-mouse (WB: 1:1000 | Jackson ImmunoResearch | RRID:AB_2340770<br>Cat: #715-035-150 |
| Peroxidase donkey anti-rabbit (WB: 1:1000) | Jackson ImmunoResearch | RRID:AB_10015282<br>Cat: #711-035-152 |
| **Oligonucleotides and other sequence-based reagents** | | |
| NDC80 gRNA sequence (5'-TCTTCACTAGAAACATCTTG-3') | Thermo Fisher | |
| MPS1 gRNA sequence (5'-TCTTTGATGCTAGTTAAAGT-3') | Thermo Fisher | |
| Control siRNA sequence (5'-CGUACGCGGAAUACUUCGAUU-3') | Dharmacon | |
| INCENP siRNA sequence (5'-GGCUUGGCCAGGUGUAUAUdTdT-3') | Dharmacon | |
| Borealin siRNA sequence (5'-AGGUAGAGUGUCUGUUCAdTdT-3') | Dharmacon | |
| Survivin siRNA sequence 5'-GCAGGUUCCUUAUCUGUCAUU-3') | Dharmacon | |
| Sgo1 siRNA sequence (5'-CAGUAGAACCUGCUCAGAA-3') | Dharmacon | |
| Borealin 10–221/280 (Forward: 5'-GAAGCTTAGCTCGGATCCATGGTGGCCAAGACCAACTCC-3') | Thermo Fisher | |
| Borealin 10–221 (Reverse: 5'-GCTCTCGAGAAGAGGGCTGCCATTCCC-3') | Thermo Fisher | |
| Borealin 10–280 (Reverse: 5'-GGGGGGGCTCGAGTTTGTG GGTCCGTATGCT-3' | **Thermo Fisher** | |
| Widom 601 147-bp sequence (5'-CTGGAGAATCCCGGTGCCGAG<br>GCCGCTCAATTGGTCGTAGACAGCTCTAGCACCGCTTAAACGC<br>ACGTACGCGCTGTCCCCCGCGTTTTAACCGCCAAGGGGATTAC<br>TCCCTAGTCTCCAGGCACGTGTCAGATATATACATCCTGT-3') | Thermo Fisher | |
| **Chemicals, enzymes and other reagents** | | |
| Puromycin | InvivoGen | Cat: # ant-pr-1 |
| Blasticidin | InvivoGen | Cat: #ant-bl-05 |
| 5-iodotubercidine (MPS1 inhibitor) | MedChemExpress | Cat: #HY-15424 |
| ZM447439 (Aurora B inhibitor) | Tocris | Cat: #2458/10 |
| RO-3306 (CDK1 inhibitor) | Tocris | Cat: #4181 |
| MG132 (Proteasome inhibitor) | Sigma-Aldrich | Cat: #474790 |
| Phos-tag Acrylamide | Fujifilm Wako | Cat: #AAL-107 |
| bis(sulfosuccinimidyl)suberate (BS3 crosslinker) | Thermo Fisher | Cat: #21580 |
| HaloTag dye JFX-646 (STED 1:10,000) | Grimm et al, 2017 | |
| Picogreen DNA dye (STED 1:1000)) | Invitrogen | Cat: # P7589 |
| **Software** | | |
| LAS X software (Version 4.8) | Leica Microsystems | |
| Fiji (ImageJ) v.2.14.0 | NIH | |
| GraphPad Prism v.10.3.0 | GraphPad Software | |
| Image Lab v.6 | Bio-Rad | |
| MATLAB 2023b | MathWorks | |
| MetaMorph 7.5 | Molecular Devices | |
| CryoSPARC v4.4.0 | Punjani et al, 2017 | |
| SIMPLE 3.0 | Caesar et al, 2020 | |
| Relion v4.0 | Kimanius et al, 2021 | |
| Chimera | Pettersen et al, 2021 | |
| DeepEMhancer | Sanchez-Garcia et al, 2021 | |
| Phenix v1.20.1 | Adams et al, 2011 | |
| Coot | Emsley et al, 2010 | |
| **Other** | | |
| Leica Stellaris STED microscope system | Leica Microsystems | |
| Titan Krios cryo-electron microscope | FEI/Thermo Fisher | |
| Talos Arctica cryo-electron microscope | FEI/Thermo Fisher | |
| Vitrobot Mark IV | FEI/Thermo Fisher | |

## Cell culture

Cell lines are validated stocks purchased from the ATCC. HeLa (#CRL-2.2) and derivatives were cultured in DMEM containing GlutaMAX™ (Gibco #10569010) and 10% [vol/vol] FBS. HCT116 cells (#CCL-247) were cultured in McCoy's 5A medium (Gibco #16600082), supplemented with 10 mM sodium pyruvate (Thermo Fisher #11360088) and 10% [vol/vol] FBS. For routine passaging, cells were washed in PBS and then incubated with TrypLE™ Express Enzyme cell dissociation reagent (Gibco #12605036) for 5 min at 37 °C, before resuspending detached cells in full medium for passage. All cell lines were maintained at 37 °C, humidified 5% $CO_2$ in a cell culture incubator (Thermo, HERAcell, #51013568). Mycoplasma negative status of cell lines was confirmed using the EZ-PCR Mycoplasma Test Kit with internal control (K1-0210, Geneflow). The cell lines used in our studies are not on the list as commonly misidentified lines.

## Generation of HCT116 NDC80-HaloTag cell lines

To generate HCT116 NDC80-HaloTag cells, a guide RNA sequence targeting the C-terminal coding exon of NDC80 was inserted into a pX459 vector was delivered together with a donor homology template encoding for 12aa (GG+2x[SG₄]) linker-HaloTag-GSG-P2A-PURO, flanked by 600 bp homology arms either side, by transient transfection. For transfection, transfection mixes were prepared in sterile DNase-free Eppendorf tubes (Eppendorf #0030108035) consisting of 100 μl Opti-MEM (Gibco #11058021), 3 μl Mirus TransIT-X2 transfection reagent (Mirus #MIR6004), and up to 1 μg of plasmid DNA. Transfection mixes were vortexed briefly for 20 s, and incubated at room temperature for 30 min, before adding dropwise to cells. Genomic DNA was isolated from clones using QuickExtract™ DNA isolation solution (Lucigen #QE09050). PCRs were performed with KOD hot-start polymerase (Merck #71086), using primers flanking the gRNA recognition site. PCR products were ligated into pSC-A vectors using a Blunt-end PCR cloning kit (Agilent #240207). Ligated plasmids were transformed into XL-1 blue competent cells, and 10 colonies were mini-prepped (Qiagen #27107) and sent for Sanger DNA sequencing (Source Genomics, Cambridge, UK) to confirm the correct sequence for the edited alleles.

## Generation of HeLa and HeLa TRex cell lines for MPS1 recruitment assays

HeLa MPS1-GFP cells expressing MPS1 endogenously tagged with GFP at the C-terminus have been described before (Alfonso-Perez et al, 2019; Hayward et al, 2019), and were used in Fig. 1. HeLa Flp-In T-REx cells (Hewitt et al, 2010) were engineered to express C-terminally GFP-tagged MPS1 created using CRISPR/Cas9. These HeLa Flp-In T-REx GFP-MPS1 cells were used in all other figures. In brief, homology recombination cassettes containing the desired knock-in DNA with flanking regions of homology of 1075 bp to the target locus were co-transfected with a version of pSpCAS9(BB) (Addgene, #48139) containing the MPS1 guide RNA sequence and modified to remove puromycin resistance. The knock-in sequences harbour a puromycin resistance marker followed by a glycine-serine rich flexible linker (GS), a P2A ribosome-skipping sequence, and the EGFP protein sequence followed by a glycine-serine rich

flexible linker (GS). Antibiotic-resistant clones were selected with 0.4 μg/ml puromycin and successful modification was confirmed by western blotting. Single integrated copies of the desired CPC transgene were introduced into HeLa Flp-In T-REx MPS1-GFP cells using the doxycycline-inducible Flp-In T-REx system (Invitrogen). The pcDNA5/FRT/TO vectors were used to make borealin and survivin constructs C-terminally tagged with mScarlet. Constructs encoding borealin and wild-type and E65A-H80A survivin have been described before (Serena et al, 2020). Borealin mutants expressing Borealin^10-280^-mScarlet and Borealin^10-221^-mScarlet- were generated using PCR to amplify the indicated regions of the wild-type sequence followed by restriction-based cloning into pCDNA5 modified to encode mScarlet at the C-terminus of the fusion protein. Antibiotic-resistant clones were selected with 4 mg/ml blasticidin and 200 mg/ml hygromycin B, and successful modification was confirmed by western blotting.

## STED Sample Preparation using HCT116 NDC80-HaloTag cell line

Cells were seeded onto high precision 22 mm 1.5H coverslips (Marienfeld #0107052). To arrest cells in G2, 6 μM of the Cdk1-inhibitor RO-3306 (Tocris #4181) was added 18 h prior to fixation. RO-3306 was washed out 4× in media containing 10 μM of the proteasome inhibitor MG132 (Sigma-Aldrich #474790, 10 mM stock) for 1–2 h to enrich for prometaphase/metaphase cells. For siRNA depletion experiments, cells were transfected with either control or Sgo1 targeting duplexes and left for 72 h. In all cases, 100 nM JFX-646 HaloTag dye was added 15 min prior to fixation to fluorescently label NDC80-Halotag (Grimm et al, 2017). Cells were then fixed for 12 min with G-PTEMF (0.1% [vol/vol] glutaraldehyde, 20 mM PIPES pH 6.8, 0.2% [vol/vol] Triton X-100, 10 mM EGTA, 1 mM $MgCl_2$, and 4% [vol/vol] formaldehyde). Fixative was quenched for 15 min with 50 mM $NH_4Cl$. Coverslips were then incubated in primary antibody for 1.5–2 h, followed by a PBS wash, and a further 1.5-hour incubation with secondary antibodies. Cells were then post-fixed for 15 min with 0.1% [vol/vol] glutaraldehyde and 3% [vol/vol] paraformaldehyde to stabilise bound antibodies. Fixative was quenched with 1 mg/ml sodium borohydride for 7 min, followed by 100 mM glycine for 15 min. Coverslips were then mounted onto cover glass using Everbrite mounting medium (Biotium #BG23001) and sealed with nail varnish. Samples were then imaged the next day using the Leica Stellaris STED system with Tausense and xTend capabilities. An NKT-Photonics 470-790 nm White Light Laser (WLL) was used to excite fluorophores, and either an MPB-589 nm or MPB-775 nm STED laser used for depletion. Images were acquired using a Leica's LAS X software (Version 4.8), alongside using a HCFL 100×/1.4 A Oil objective, a HydX detector, and an average z-spacing of 0.3–0.4 μm.

## Image processing and data analysis for STED images

Image analysis was performed using the Fiji distribution of ImageJ (V2.14). Quantification was performed on a per kinetochore pair basis (either on a single slice or sometimes needing a maximum projection of $x$ number of slices). Line scans were generated by measuring the signal spanning the pericentric region between two kinetochore pairs. The same line generated was used to measure density across all channels. The pixel distances were converted to

µm dependent on the scaling of each image when acquired. This data was then normalised in GraphPad Prism Version 10 for MacOS, with 0 defined as the smallest value in each data set, and 100% defined as the largest mean in each data set. Each scan was plotted with ± SEM, shown as a lightly coloured region about the curve. Inter-kinetochore distances were measured using a 1 px wide line in Fiji, with each end placed centrally about the kinetochore. For these quantifications, a single pair of kinetochores was considered a replicate within a cell (*n*). Intra-kinetochore distances were determined by measuring a line plot across individual kinetochores and fitting a Gaussian to each signal. The difference between means of corresponding CENP-A and NDC80 gaussians was used as a measure of distance between them, due to the high resolution of the image and sharpness of gaussian fits. For these quantifications, a single kinetochore was considered a replicate within a cell (*n*). Statistical analysis of these distances was performed using a parametric two-tailed unpaired *t* test. Graphs display the mean ± SD, with *P* values shown on the graphs as follows: $P > 0.05$ (not significant, n.s.), $P < 0.05$ (*), $P < 0.01$ (**), $P < 0.001$ (***), $P < 0.0001$ (****).

## MPS1 recruitment assays

To test the role of CPC subunits in MPS1 recruitment and histone H3 S10-phosphorylation, HeLa cells or HeLa Flp-In T-REx cells expressing endogenously C-terminally tagged MPS1 were depleted of specific CPC subunits and arrested in mitosis. Cells were grown on coverslips (16 mm circle No. 1 ½, Thermo Fisher) placed in six-well plates a day before treatment. All siRNA depletions were performed for 48 h using Control siRNA or siRNA targeting the 3'-UTR of the CPC components INCENP, Survivin and Borealin, or the accessory factor Sgo1. For rescue experiments using HeLa MPS1-GFP Flp-In T-Rex borealin- or surviving-mScarlet, 2 µM doxycycline was added 30 min before siRNA addition to induce mScarlet-transgene expression. A second doxycycline induction was performed 24 h into siRNA depletion.

To arrest the cells in a spindle checkpoint active prometaphase-like state and depolymerise microtubules, 0.66 µM nocodazole was added for 2 h. To prevent mitotic exit in cases where the spindle checkpoint was compromised, 20 µM of the proteasome inhibitor MG132 was added 30 min before fixation. For spindle checkpoint re-activation assays (Fig. 2E–G), cells were treated with 20 µM MG132 for 90 min to cause mitotic arrest with spindle checkpoint-silenced metaphase plates, followed by either Haspin or Aurora B inhibition using 5-iodotubercidin (5-iTu, 10 µM) and ZM447439 (10 µM) for 10 min, respectively, followed by a high dose of 6.6 µM nocodazole for 5 min to rapidly activate the spindle assembly checkpoint. Cells were fixed with PTEMF buffer (20 mM PIPES-KOH pH 6.8, 0.2% [vol/vol] Triton X-100, 10 mM EDTA, 1 mM MgCl₂, 4% [vol/vol] formaldehyde) for 12 min at room temperature. Coverslips were incubated in blocking buffer (3% [wt/vol] bovine serum albumin) for a minimum of 1 h. Coverslips were incubated face-down on 80 µl droplets of primary antibodies in a humidified chamber for 1 h. Following primary antibody incubation coverslips were washed 3× in PBS. Coverslips were incubated face-down on 80 µl droplets of diluted secondary antibodies in a humidified chamber for 45 min. Coverslips were washed 3× in PBS and 1× in distilled deionised water. Coverslips were left to dry completely before being mounted with 7 µl of Mowiol 4-88 (Sigma)

according to manufacturer's instructions. Samples were imaged using a 603/1.35-NA oil-immersion objective on a BX61 Olympus microscope equipped with filter sets for DAPI, EGFP/Alexa Fluor 488, 555 and 647 (Chroma Technology), a CoolSNAP HQ2 camera (Roper Scientific), and MetaMorph 7.5 imaging software (Molecular Devices). Each image consisted of a total stack of 2 µm with 0.2-µm intervals.

## Image processing and data analysis for MPS1 recruitment assays

Image processing was performed using the FIJI distribution of ImageJ or MATLAB 2023b (MathWorks, Natick, MA, USA). Quantification was performed on sum-projected images whilst figures shown were collated from maximum projections. Individual cells were cropped to 250 × 250-pixel images. The quantitation of MPS1 and BubR1 kinetochore signals was performed on FIJI, as previously described in Hayward et al (2022). Kinetochore intensities for each fluorescence channel were determined by placing 8 px-diameter circular regions of interest (ROIs) at the maxima of individual non-overlapping kinetochores and measuring the mean pixel intensity of each channel within said selections. A total of 20 kinetochores were measured per cell. Background measurements were derived by taking an equivalent number of pixels as were in the ROI which were as close as possible to the ROI without overlapping with kinetochores. In brief, a binary mask of kinetochore signal was generated by performing a top-hat transform of the CENP-C channel and thresholding using an iterative intermeans method. Pixels were radially selected from outside the kinetochore ROI and, if not overlapping with signal in the binary kinetochore mask, added to a new background ROI. Once 52px had been incorporated into the background ROI the mean pixel intensity of each channel within said ROI was measured. Using an automated script on MATLAB 2023b, kinetochore signal intensities were background-adjusted by subtracting the background signal on a channel-by-channel basis. Any negative values were set to zero. The mean intensity of the channel of interest was then divided by the mean intensity of the CENP-C channel on a per-kinetochore basis. The mean kinetochore localisation intensities were then calculated for each cell. Quantitation of pericentromeric CPC subunits and Aurora B, and chromatin-associated Histone H3 pSer10 (H3pS10) signal was performed on MATLAB 2023b. In brief, binary masks were defined for the chromatin using Hoechst staining and the cytoplasm for each cell. The mean fluorescence signals of CPC/Aurora B and H3pS10 were then measured, and in the case of CPC/Aurora B, background-corrected by subtracting the mean cytoplasmic signal. Any negative values were set to zero. Finally, the measurements were normalised by the mean DNA signal detected using Hoechst fluorescence. For graph plotting and statistical analysis, all measurements were scaled to the mean fluorescence signal of the control condition, either the control depletion or control treatment in Fig. 2 and Appendix Fig. S2 or the borealin or survivin wild-type rescue conditions in Figs. 4 and 5, respectively.

## Statistical analysis for MPS1 recruitment assays

All statistical analysis was performed using GraphPad Prism Version 10 (GraphPad Software, San Diego, CA, USA). Each cell

measured was considered as a biological replicate (*n*), hence mean measurements calculated for each cell were used for statistical analysis. At least three independent repeats of each experiment were performed, with statistical analysis performed using a sum of biological replicates from all independent experiments. The precise n numbers (where n is the number of cells analysed) is indicated in the figures. For CPC subunit depletion experiments in Fig. 2 and Appendix Fig. S2, experimental conditions were compared to the control siRNA. For rescue experiments with borealin and survivin or mutants thereof in Figs. 4 and 5, wild-type rescue conditions were compared to the "no rescue" control, and mutants were then compared to the wild-type rescue condition. A non-parametric Kruskal–Wallis test with Dunn's test for multiple comparisons was used for the analysis. Graphs display the mean ± SEM and individual data points for each cell. The *P* values shown on the graphs are as follows: $P > 0.05$ (not significant, ns), $P < 0.05$ (*), $P < 0.01$ (**), $P < 0.001$ (***), $P < 0.0001$ (****).

## Protein expression

### Chromosomal passenger complex production

Human CPC was expressed using the pETDuet-1 vector system engineered to contain survivin (full-length, residues 1–142) with an N-terminal 6x-His tag and an HRV 3C protease site between its NcoI and HindIII restriction sites, as well as human borealin between its NdeI and XhoI sites (either borealin$_{(1-280)}$ or borealin$_{(1-76)}$, including the R9A, K12A, and R9A-K12A mutants). Human INCENP$_{(1-80)}$ was cloned into a pFAT vector between NcoI and XhoI restriction sites, with a 6x-His and GST tags at its N-terminus followed by thrombin and HRV 3C cleavage sites. Both pETDuet-1 and pFAT vectors containing genes of interest were used to transform BL21(DE3) pLysS or BL21-CodonPlus(DE3)-RIL *E. coli* cells, respectively. The transformed cells were plated on agar containing 100 µg/ml ampicillin and 34 µl/ml chloramphenicol, and single colonies were used to inoculate 2xYT media supplemented with antibiotics. Cultures were grown at 37 °C with 180 rpm shaking, and protein expression was induced at 0.6 OD$_{600}$ with 0.5 mM IPTG, after overnight induction at 18 °C the cells were harvested by centrifugation at $6000 \times g$. The combined survivin/borealin and INCENP pellets were mixed at a 2:1 [wt/wt] ratio, and the mixture was resuspended in CPC buffer (20 mM Tris-HCl, pH 7.5, 500 mM NaCl, 5 mM DTT) supplemented with protease inhibitors (Roche). The suspension was then sonicated and centrifuged at $30,000 \times g$ for 45 min. The supernatant was applied to a 5 ml GSTrap column (Cytiva) using a flow rate of 1 ml/min. The column was then washed with CPC buffer until A$_{280}$ reading was stabilised, and the protein eluted using 25 mM glutathione in CPC buffer. The protein concentration of the elution was measured, and GST-tagged HRV 3C protease was added at a 1:150 molar ratio. The sample was then dialysed against fresh CPC buffer using 7 kDa MWCO SnakeSkin dialysis tubing (Thermo Scientific) overnight at 4 °C. Next day the sample was reverse purified by passing it over a 5 ml GSTrap column, concentrated using a 10 kDa MWCO protein concentrator (Cytiva), and injected on an S200 10/300 GL column (GE). The eluted fractions were analysed by SDS-PAGE, and the peak containing both survivin, borealin, and INCENP was pooled and concentrated. When constructs containing borealin$_{(1-76)}$ were purified, the final sample was supplemented with 10% [vol/vol] glycerol, flash-frozen in

liquid nitrogen, and stored at −80 °C. With the wild-type borealin$_{(1-280)}$ construct it was only used fresh.

### Histone preparation

BL21(DE3) pLysS *E. coli* cells were transformed with pET3-a vectors containing histone genes (*X. laevis* H2A, H2B, H3, H4) between the NdeI and BamHI restriction sites of the vector (Luger et al, 1997a, b). Transformed cells were plated on agar containing 100 µg/ml ampicillin and 34 µl/ml chloramphenicol. Fresh colonies were used to grow cultures in 2xYT media supplemented with antibiotics, at 37 °C, 180 rpm. Protein expression was induced by the addition of 0.2 mM IPTG once OD$_{600}$ of cell cultures had reached 0.6. Cells were harvested after 3 h by centrifugation at $6000 \times g$, and the pellets resuspended in histone wash buffer (50 mM Tris-HCl pH 7.5, 100 mM NaCl, 1 mM EDTA) supplemented with protease inhibitors (Roche). The pellets were extensively sonicated, followed by centrifugation at $30,000 \times g$ for 45 min. The pellet was then washed two times, resuspending the pellet in histone wash buffer supplemented with 1% [vol/vol] Triton X-100 and repeated centrifugation, followed by two additional washes without Triton X-100 present. Subsequently, the inclusion body pellet was incubated in 0.5 ml DMSO for 30 min, after which 2 ml of histone unfolding buffer (7 M guanidine-HCl, 20 mM Tris-HCl pH 7.5, 10 mM DTT) was added and the pellet incubated for an additional hour with agitation. The suspension was then centrifuged for 20 min at $20,000 \times g$, and the supernatant loaded on a Sephacryl S200 26/60 column (GE) equilibrated with SAU buffer (7 M urea, 20 mM NaOAc pH 5.2, 1 M NaCl, 1 mM EDTA, 5 mM β-mercaptoethanol). Elution fractions were analysed on an SDS-PAGE gel, and the ones containing histone were pooled, dialysed against water, and dried. The protein pellets were resuspended in SAU buffer containing 200 mM NaCl and loaded on a 5 ml Resource S cation exchange column (Cytiva). Fractions were eluted by increasing the salt concentration in a stepwise fashion using SAU buffer containing 600 mM NaCl. The fractions were again analysed on an SDS-PAGE gel, and the ones containing histones were pooled, dialysed against water, dried, and stored at −80 °C.

## T3-phosphorylation of histone H3 with a recombinant Haspin kinase domain

The kinase domain of Haspin was expressed from a pNIC28-Bsa4 vector containing human Haspin residues 465–798 (Addgene plasmid #38915, received as a gift from Nicola Burgess-Brown). The plasmid was used to transform BL21(DE3) pLysS *E. coli* cells, which were plated on agar containing 50 µg/ml kanamycin and 34 µl/ml chloramphenicol. Next day, a single colony was used to inoculate 2xYT media supplemented with antibiotics, and the culture was grown at 37 °C, 180 rpm. Protein expression was induced at 0.6 OD$_{600}$ with 1 mM IPTG overnight at 18 °C, after which the cells were harvested by centrifugation at $6000 \times g$ and the pellet resuspended in Haspin buffer (30 mM HEPES, pH 7.5, 500 mM NaCl, 5% [vol/vol] glycerol, 1 mM DTT) supplemented with protease inhibitors (Roche) and 5 mM imidazole. The cells were then lysed by sonication, and the lysate loaded on a 5 ml HisTrap HP column (Cytiva) equilibrated with the same buffer. Fractions were collected while increasing the imidazole concentration in a stepwise manner. Following analysis by SDS-PAGE, the

fraction containing Haspin (eluting at 250 mM imidazole) was collected, concentrated using a 10 kDa MWCO concentrator (Vivaspin) and loaded on a S200 10/300 GL column (GE) equilibrated with Haspin buffer. The eluted fractions were analysed by SDS-PAGE, and the peak containing haspin was pooled, concentrated, supplemented with 10% [vol/vol] glycerol, flash-frozen in liquid nitrogen, and stored at −80 °C.

Phosphorylation of H3 was performed by resuspending dry H3 pellets in 25 mM HEPES-NaOH pH 7.5, 5 mM MgCl$_2$, 300 μM ATP, 100 mM NaCl, 5 mM DTT to a final concentration of 2 mg/ml. Recombinant Haspin kinase domain was then added to the required concentration (for routine phosphorylation—1.3 μM, 1:100 Haspin:H3 molar ratio) and the mixture incubated at 4 °C overnight. For analysis by Phos-tag-SDS-PAGE, the 15% [wt/vol] SDS-PAGE gels were cast by including 50 μM Phos-tag acrylamide (Fujifilm Wako) and 5 mM MgCl$_2$ in the gel mixture. Western blotting was performed using Trans-Blot Turbo 0.2 μm nitrocellulose transfer packs (Bio-Rad), 5% [wt/vol] skim milk powder, PBS-0.1% [vol/vol] Tween 20 using antibodies for histone H3 and T3-phosphorylated histone H3.

## Assembly of H3T3 and H3pT3 nucleosomes

Dried histones were resuspended in histone unfolding buffer to a concentration of 2 mg/ml and agitating them for 1 h. The histone mixtures were then combined at an equimolar ratio and diluted to a 1 mg/ml concentration using the same buffer. If phosphorylated H3 was used, it was added to the other histones immediately after Haspin treatment. The histone mixture was then dialysed against histone refolding buffer (10 mM Tris-HCl, pH 7.5, 2 M NaCl, 1 mM EDTA, 5 mM β-mercaptoethanol) at 4 °C using 3 kDa MWCO SnakeSkin dialysis tubing (Thermo Scientific). The sample was then concentrated using a 10 kDa MWCO protein concentrator (VivaSpin) and loaded on a S200 10/30 GL column (GE) equilibrated with histone refolding buffer. The eluted fractions were analysed by SDS-PAGE, the main peak containing the histone octamer was collected, concentrated using a 30 kDa MWCO protein concentrator (VivaSpin), and stored at 4 °C. To produce the nucleosomal DNA, 16 copies of the Widom 601 147-bp core sequence were cloned into a pUC19 vector following a strategy described by (Dyer et al, 2004). In the final plasmid the copies of 601 DNA were located between EcoRV restriction sites, which, following cleavage, would result in additional 3 or 2 base pairs at the 5' and 3' ends of the 601 sequence respectively. The plasmid was used to transform XL1-Blue competent cells, which were plated on agar containing 100 μg/ml ampicillin. Next day, a single colony was used to start a culture in TB media supplemented with antibiotics, which was grown at 37 °C, 180 rpm. Upon reaching an OD$_{600}$ of 0.6, the cells were grown for additional 15 h and harvested by centrifugation at 6000 × g for 45 min. The plasmid was isolated from the resulting cell pellets by using a Plasmid Mega Kit (Qiagen), adjusting the starting material to 1 g of cell pellet per each plasmid purification column. The plasmid was then restriction-digested overnight at 37 °C with EcoRV-HF (NEB), using 1 kU of enzyme per nmol of plasmid. The mixture was then loaded on 2 ×1 ml HiTrap Q columns (Cytiva) equilibrated with IEX buffer A (20 mM Tris-HCl, pH 7.5, 100 mM NaCl, 1 mM EDTA), and anion exchange chromatography performed at 0.5 ml/min, first by applying a gradient of IEX buffer B (20 mM Tris-HCl, pH 7.5,

1000 mM NaCl, 1 mM EDTA) to 55% over 12 min, followed by a gradient to 75% over 80 min. Peak fractions were analysed on a 1% [wt/vol] agarose gel, and the peak containing 601 DNA was pooled, ethanol-precipitated, and stored at −20 °C. The Widom 601 DNA to NCP octamer ratio for each batch of nucleosomes was determined by small-scale reconstitutions with different DNA:NCP ratios ranging from 0.80 to 1.05 (mol:mol). Each reaction was set up using Widom 601 DNA at a 5.5 μM concentration, with the final buffer composition adjusted to be the same as the histone refolding buffer. The NaCl concentration within the reconstitution mixture was then gradually reduced to 100 mM by the addition of 10 mM Tris-HCl pH 7.5, over the course of 6 h. The final mixtures were analysed using a 0.8% agarose gel run at room temperature, 90 V (constant voltage), 50 min. The ratio for large-scale reconstitutions was selected based on the intensity of the band with a migration distance matching the 1.5 kbp marker. Large-scale NCP reconstitution was performed the same way as previously, using more starting material and the selected 601 DNA to NCP octamer ratio. The final mixtures were injected on 2 ×1 ml HiTrap Q columns (Cytiva) equilibrated with IEX buffer A (20 mM Tris-HCl, pH 7.5, 100 mM NaCl, 1 mM EDTA), and anion exchange chromatography performed at 0.5 ml/min, first by applying a gradient of IEX buffer B (20 mM Tris-HCl, pH 7.5, 1000 mM NaCl, 1 mM EDTA) to 45% [vol/vol] over 12 min, followed by a gradient to 65% over 90 min. Peak fractions were analysed on a 0.8% [wt/vol] agarose gel, and the peak exhibiting the slowest-migrating bands was pooled, dialysed against 10 mM Tris-HCl, pH 7.5, 5 mM DTT, and stored at 4 °C.

## Electrophoretic mobility shift assays (EMSAs)

Nucleosome samples were supplemented with buffer to yield a final concentration of 150 nM and a buffer composition corresponding to complex buffer (10 mM Tris-HCl, pH 7.5, 150 mM NaCl, 2 mM DTT). CPC samples were dialysed against the same buffer and added to nucleosomes to yield the required nucleosome to CPC ratio. The mixtures were incubated on ice for 1 h, while Mini-Protean TBE precast 5% polyacrylamide gels (Bio-Rad) were pre-run for 15 min, 90 V (constant voltage), 4 °C, using 0.5× TBE buffer (45 mM Tris-borate, pH 8.3, 1 mM EDTA). The mixtures were then loaded on the pre-run gels, and run in 0.5× TBE buffer, 4 °C, 20 V (constant voltage), 45 min, after which the voltage was increased to 90 V and the run continued for additional 45 min. The gels were stained in 0.5x TBE supplemented with 1× SYBR Gold (Invitrogen) for 20 min and immediately imaged. All gels intended for comparison were run simultaneously.

## Cryo-electron microscopy sample preparation and data collection

CPC80-borealin$_{(1-76)}$ samples were dialysed against the complex buffer and added to H3pT3-NCP, with the sample containing a 7× molar excess of CPC and a final buffer composition corresponding to the complex buffer. The samples were then incubated for 1 h on ice, supplemented with 0.3% n-octyl-β-D-glucoside (βOG, Avanti Polar Lipids), and immediately used for grid preparation. Samples containing CPC80-borealin$_{(1-280)}$ were dialysed against 20 mM HEPES (pH 7.5), 500 mM NaCl, 1 mM DTT, and added at a 4× molar excess to 2.3 μM H3pT3-NCP in the same buffer. The

samples were incubated on ice for 1 h, supplemented with bis(sulfosuccinimidyl)suberate (BS3, Thermo Fisher) at a 1:6 (protein:BS3) mass ratio, incubated for additional 2 h on ice, and quenched by the addition of 50 mM Tris-HCl. The sample was then dialysed against the complex buffer, supplemented with 0.3% β-octylglucoside and immediately used for grid preparation. A 3.5 µl sample at 10 µM concentration was applied to QuantiFoil 1.2/1.3 grids (200 mesh) glow-discharged at 15 mA for 25 s using Leica EM ACE200. The grids were blotted and plunge-frozen in liquid ethane using Vitrobot Mark IV (FEI) operated at 4 °C, 100% humidity, −10 blot force, and 3 s blotting time. The grids were clipped and screened for suitable particle distribution and ice thickness using an FEI Talos Arctica cryo-electron microscope operating at 200 kV and equipped with a Falcon 4 direct electron detector (Thermo Scientific). Preliminary data collection was performed in counting mode, using a pixel size of either 1.2 or 1.5 Å/px.

Full datasets from selected grids were collected using an FEI Titan Krios, operating at 300 kV, equipped with a K3 direct electron detector (Gatan) and a BioQuantum energy filter (20 eV). Dataset acquisition was automated using the fast acquisition mode in EPU (Thermo Scientific). For the H3pT3 nucleosome:CPC80-borealin$_{(1-76)}$ dataset, movies were acquired in 2×-binned super-resolution mode with a nominal magnification of 58,009× and a calibrated pixel size of 0.830 Å/px. The dose rate was 12.038 e$^-$/px/s, with a total dose of 43.860 e$^-$/Å$^2$ over 2.51 s and 40 frames. A defocus range of −1.0 to −2.3 µm was applied, and a 100 µM objective aperture was used. In total 20,239 movies were collected in the same session, from a single grid. For data collection details relating to other datasets, please consult Table EV1.

## Cryo-electron microscopy data processing

### H3pT3 nucleosome:CPC80-borealin$_{(1-76)}$ complex

Movies were motion-corrected on-the-fly using SIMPLE 3.0 (Caesar et al, 2020). The corrected micrographs were then exported to CryoSPARC v4.4.0 (Punjani et al, 2017) and their CTF calculated using patch CTF estimation. Micrographs containing poor CTF parameters, broken regions, or protein aggregation were manually discarded, leaving 19,452 micrographs. A subset of 12,458 micrographs was used for blob-based particle picking, setting the expected particle diameter to 110 Å. The resulting 1,201,581 particles were extracted with a box size of 260 px and subjected to 2D classification using a circular mask diameter of 170 Å. Well-defined classes were then used as templates for template-based picking of all micrographs. The resulting 1,946,286 particles were extracted with a box size of 384 px and Fourier-cropped to a box size of 96 px. Junk particles were discarded by performing three rounds of 2D-classification, followed by particle re-extraction without Fourier cropping and one final 2D classification. Since all 2D classes clearly contained H3pT3 nucleosome, all 1,674,695 clean particles were used to reconstruct a single ab initio class, which corresponded to the structure of the H3pT3 nucleosome, with an additional density with much weaker signal corresponding to bound CPC. The ab initio model was then used in a heterogenous refinement with two classes. The class with the better-defined density containing 70.5% of the particles was then subjected to non-uniform (NU) refinement with per-particle defocus and CTF parameter optimisation. It yielded a map at 2.3 Å gold-standard FSC (GSFSC) resolution. The map contained a well-defined H3pT3 nucleosome density, with a relatively small region of additional

density on its surface, which, based on the fitting of previously published CPC crystal structures (PDB ID: 6YIH) in the best heterogenous refinement class, corresponded to the N-terminus of borealin.

All particles used for calculation of the ab initio model were used in a new homogenous refinement, and the CPC crystal structure (PDB ID: 6YIH) fitted in the region corresponding to CPC. It was then used to create a mask with a soft padding of 13 Å and used in a soft, focused 3D classification with ten classes and a 6 Å target resolution using the particle alignments from the previously performed homogenous refinement. The results showed CPC density in a variety of different orientations relative to H3pT3 nucleosome, as well as classes where CPC density was extremely poor. The particle distribution between classes was equal, ranging between 9.1 and 11.1%. The classes showing major differences in CPC orientation (classes 2, 4, 5, 6, 682,233 particles total) were selected for 3D variability analysis. A random subset of 113,705 particles from these classes were Fourier-cropped to a box size of 192 px and used to perform a homogenous refinement (reaching 3.41 Å GSFSC resolution), which was then subjected to localised 3D variability analysis with default parameters, using the same mask that was previously used for 3D classification. Class 6 from 3D classification containing 166,715 particles was selected for further processing due to it exhibiting the highest local resolution in the regions of the map corresponding to CPC. It was subjected to a round of heterogenous refinement with two classes, and the best class (121,922 particles, 73.1%) subsequently used in a NU-refinement with per-particle defocus and CTF parameter optimisation, yielding a 2.70 Å GSFSC resolution map. The particles were exported to Relion v4.0 (Kimanius et al, 2021), subjected to bayesian polishing with default parameters, and reimported in Cryosparc V4.4.0. Another round of NU-refinement produced a map at 2.4 Å GSFSC resolution reconstructed from 119,063 particles. Further rounds of CTF parameter optimisation or Bayesian polishing did not lead to further improvements in map quality.

This map was then used to create a mask in Chimera (Pettersen et al, 2021) that corresponds to the distal half of CPC (containing the survivin BIR domain), as well as the corresponding inverse mask. Both masks were imported in CryoSPARC, modified with a 33 Å soft padding, and the inverse mask used to perform particle subtraction with default parameters. The subtracted particles were subjected to local refinement with the CPC distal end mask, which produced a 6.87 Å GSFSC resolution map. This was sharpened with the DeepEMhancer wide target model (Sanchez-Garcia et al, 2021), and combined with the previously obtained 2.4 Å GSFSC map in Chimera v1.14.

### BS3-crosslinked H3pT3-NCP and CPC80-borealin$_{(1-280)}$ complex

In total, 3042 movies were motion-corrected on-the-fly using SIMPLE 3.0 and exported to CryoSPARC v4.4.0. CTF parameters were calculated using patch CTF estimation, and the quality of micrographs manually inspected, leaving 2856 exposures. A subset of 1000 micrographs was used for blob-based particle picking, with the expected particle diameter range set to 80–200 Å, yielding 268,699 particles. They were extracted with a 256 px box size and subjected to 2D classification, and the best classes selected for template-based particle picking. The resulting 1,287,276 particles were extracted with a box size of 256 px and used for multiple

rounds of 2D classification to get rid of particles deemed to be of low quality. The remaining 335,904 particles were used to construct 3 ab initio models, out of which two resembled the nucleosome (247,764 particles, 73.8%). These particles were used to set up a heterogenous refinement, using the best and the worst ab initio models (the latter acting as a decoy class). The best class from this job (146,173 particles, 59.0%) was used to perform a NU-refinement, which produced a map at 4.3 Å GSFSC resolution. Any additional processing steps performed did not lead to any further improvements in map quality.

### H3pT3-NCP with and without BS3 crosslinking

Datasets for both BS3-crosslinked and not crosslinked H3pT3-NCP (3222 and 2278 movies respectively) were processed in a similar fashion, with all steps performed in CryoSPARC v.4.4.0. After full-frame motion correction and patch CTF estimation, the micrographs were manually curated and a 500 micrographs subset used for blob-based particle picking, with a particle diameter range set to 90–120 Å. The particles were extracted with either a 200 or 168 px box size (for crosslinked and not crosslinked datasets, respectively) and used for 2D classification. The best classes were selected for template-based particle picking, which yielded either 1,051,059 or 491,041 particles (crosslinked and not crosslinked). These were extracted with the same box sizes as before, subjected to multiple rounds of 2D classification, and then used for ab initio model generation with two classes. The best class was then used to perform homogenous refinement. For the crosslinked dataset, this led to a final map with a GSFSC resolution of 6.7 Å and 102,960 particles, whereas for the crosslinked dataset the final resolution was 7.6 Å and 115,357 particles.

### Structural modelling

The nucleosome within the obtained maps was modelled by using the human nucleosome structure (PDB ID: 3AFA). It was rigid-body fitted within the initial 2.3 Å map containing H3pT3 nucleosome and the borealin N-terminus using Phenix v1.20.1 (Adams et al, 2011), the residues differing between the human and *Xenopus* nucleosomes manually mutated and fitted within Coot (Emsley et al, 2010), and the structure then subjected to real-space refinement in Phenix. The density of the DNA bases within the map was sufficiently well-defined to identify them as either purines or pyrimidines, and this information was used to model the exact location of 601 DNA sequence within the final map. The histone tail residues absent in the starting model were modelled if permitted by the density. The borealin N-terminus was modelled manually in Coot, up to residue 23, after which the density was absent. The structure was then subjected to several rounds of real-space refinement in Phenix followed by manual modelling in Coot to improve model geometry. The map of the CPC distal end was used to rigid body fit the previously reported human CPC crystal structure with H3pT3 peptide (PDB ID: 6YIH, containing borealin$_{(17-76)}$, survivin$_{(4-138)}$, INCENP$_{(7-43)}$, and first seven residues of the phosphorylated peptide). The residues at the H3pT3 nucleosome-proximal end were trimmed due to their density being absent, and the N-terminus of INCENP extended by two residues. The final model using this map thus contained borealin$_{(28-76)}$, survivin$_{(4-136)}$, INCENP$_{(5-39)}$, and the H3pT3 peptide. The structure was then subjected to several rounds of real-space refinement in

Phenix with reference model restraints based on 6YIH, followed by manual modelling in Coot to improve model geometry. Both the H3pT3 nucleosome and truncated CPC models were rigid body fitted into the map formed by combining the 2.4 Å refined 3D classification map and the CPC localised refinement maps. The missing residues of CPC were modelled in Coot, both models fused, and subjected to several rounds of real-space refinement in Phenix and manual modelling in Coot, while fixing the regions of the structure corresponding to the local refinement model. The final model was then also fitted into the original 2.4 Å map without further changes. The potential paths of both H3 tails interacting with CPC were modelled manually in Coot, using the combined cryo-EM map. Even though there was no density present for H3 residues 8–36, we placed them in a way that they would take the shortest path possible to the H3 N-terminus visible in our model while keeping the residue geometry sensible. This was done to see whether it was possible for each tail to reach their binding site in CPC. If it was, we positioned the rest of the tail so that all the H3 residues could be modelled between positions 7 and 37 already present in the structure.

## Data availability

Source data are BioStudies https://doi.org/10.6019/S-BSST2028 accession S-BSST2028. Molecular structures are deposited in the PBD. The Chromosomal Passenger Complex (CPC) localization module in complex with a H3pT3 nucleosome is under deposition D_1292134922 and assigned the following accession codes: PDB ID 8RUP, EMD-19513. The borealin N-terminus in complex with a H3pT3 nucleosome is under deposition D_1292134988 and assigned the following accession codes: PDB ID 8RUQ, EMD-19514. Depositions D_1292136727 and D_1292136726, relate to EMD-19685, the refinement after 3D classification (Table EV1B) supporting Fig. 2, and EMD-19684, the refinement focused on the BIR domain end of CPC (Table EV1C) shown in Fig. 4B, respectively.

The source data of this paper are collected in the following database record: biostudies:S-SCDT-10_1038-S44319-025-00523-4.

## Peer review information

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

## Acknowledgements

We thank Luke Lavis for the exceptionally generous gift of Halo-JFX dyes, our colleagues in the Barr group for discussion and advice, the COSMIC cryo-EM facility for their help with collection and processing of data, and the Oxford Leica Microsystems Centre of Excellence at the Micron Bioimaging Facility for their support & assistance in this work. This research was supported by Cancer Research UK program grant awards C20079/A24743 (FAB), DRCRPG-May23/100006 (FAB) and DRCNPG-Nov21\100004 (UG), and MRC PhD studentships to (DBHG and ER).

## Author contributions

**Reinis R Ruza**: Conceptualization; Data curation; Formal analysis; Investigation; Methodology; Writing—original draft; Writing—review and editing. **Chyi Wei Chung**: Formal analysis; Investigation; Writing—review and editing. **Danny B H Gold**: Data curation; Formal analysis; Investigation; Methodology; Writing—review and editing. **Michela Serena**: Investigation; Writing—review and editing. **Emile Roberts**: Methodology; Writing—review and editing. **Ulrike Gruneberg**: Supervision; Funding acquisition; Writing—review and editing. **Francis A Barr**: Conceptualization; Data curation; Formal analysis; Supervision; Funding acquisition; Writing—original draft; Project administration; Writing—review and editing.

Source data underlying figure panels in this paper may have individual authorship assigned. Where available, figure panel/source data authorship is listed in the following database record: biostudies:S-SCDT-10_1038-S44319-025-00523-4.

## Disclosure and competing interests statement

The authors declare no competing interests.

# Expanded View Figures

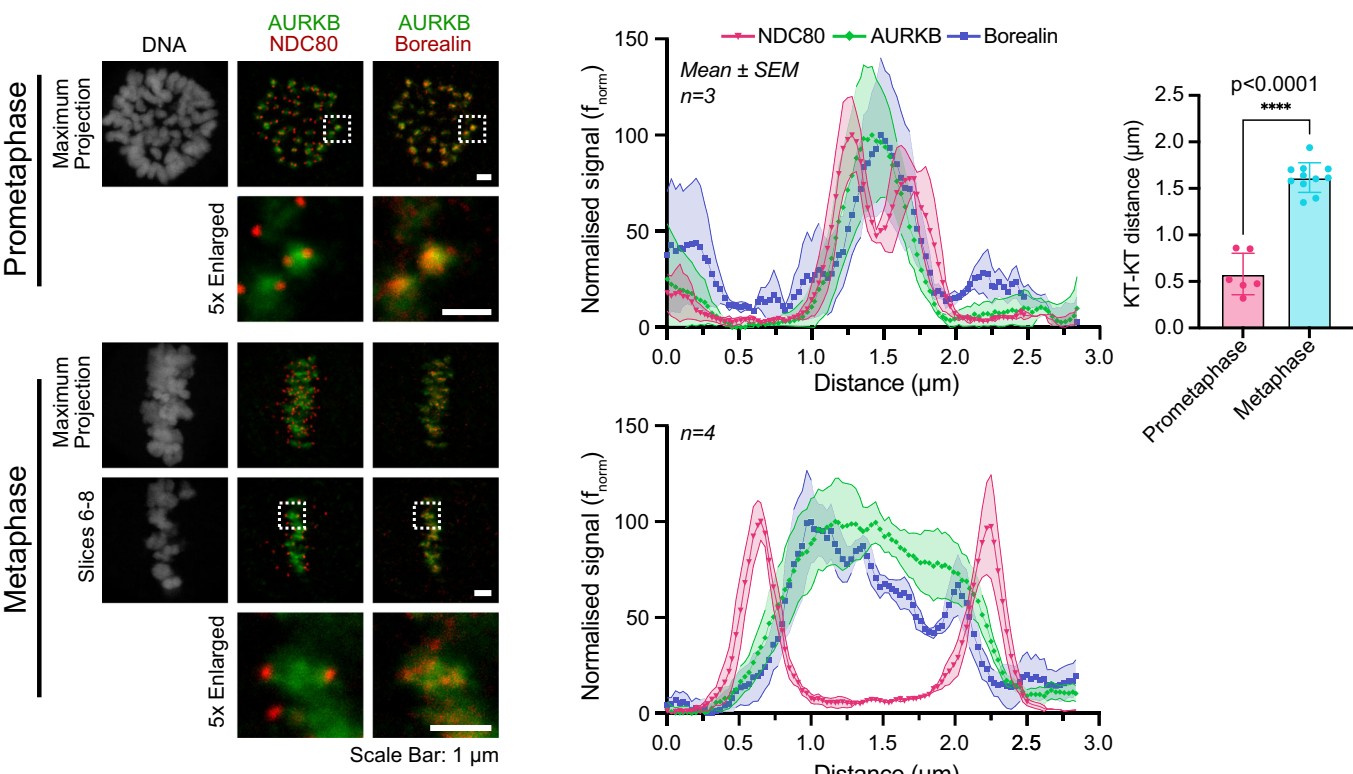

**Figure EV1.   The outer kinetochore and CPCborealin are spatially separated in metaphase.**

STED images depicting NDC80, borealin and Aurora B (AURKB) localisation in prometaphase and metaphase HCT116 NDC80-HaloTag cells. DNA was stained with Picogreen. Representative maximum intensity projections and selected slices are shown (mean ± SEM, sample size $n = 3$ or 4 as indicated in the figure). Line scans show signal intensity across kinetochore pairs. Bar graphs show mean kinetochore-kinetochore (KT-KT) distances in prometaphase and metaphase (mean ± SD). For KT-KT distances prometaphase $n = 6$, metaphase $n = 11$ with a two-tailed unpaired $t$ test, $P < 0.0001$ (****).

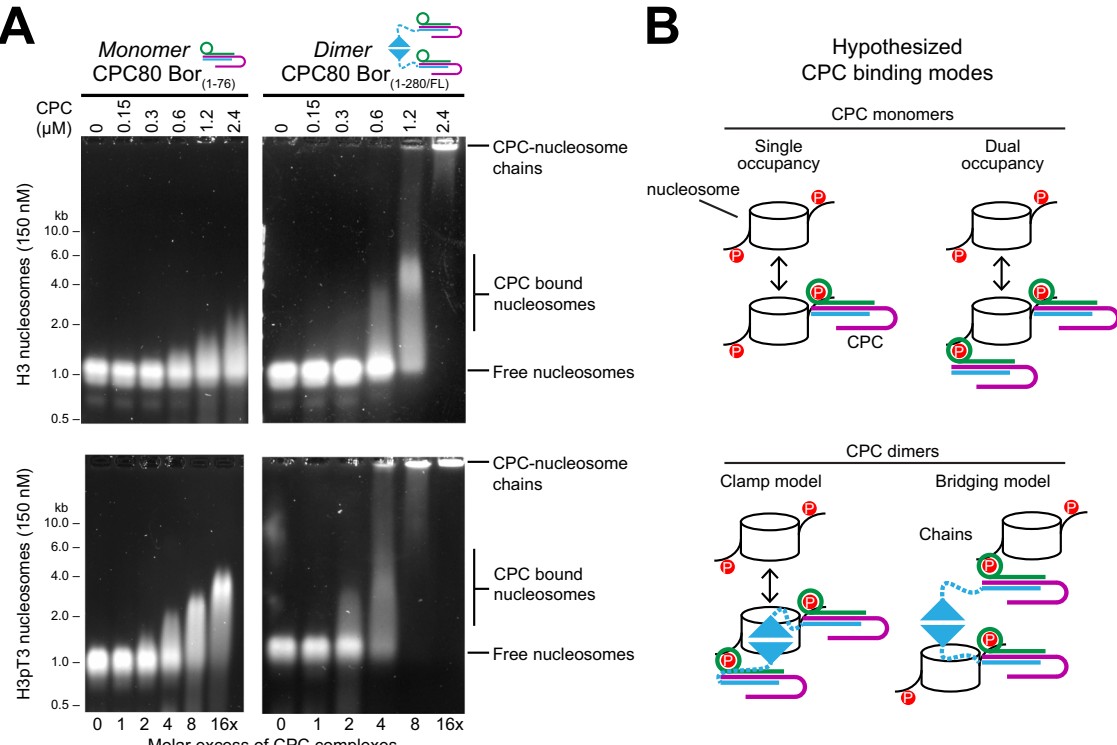

**Figure EV2. Interaction of monomeric and dimeric CPC complexes with H3pT3 nucleosomes.**

(A) EMSAs between 150 nM H3 T3-phosphorylated (H3pT3) nucleosomes or non-phosphorylated H3-nucleosomes and CPC80-Bor$_{(1-76)}$ or CPC80-Bor$_{(FL)}$ targeting module complexes at the specified molar excesses. Predicted species are indicated on the right of the gel panels. (B) A schematic depicting hypothesized binding modes for monomeric and dimeric forms of the CPC and histone H3pT3 nucleosomes (survivin in green; borealin in light blue, INCENP in magenta).

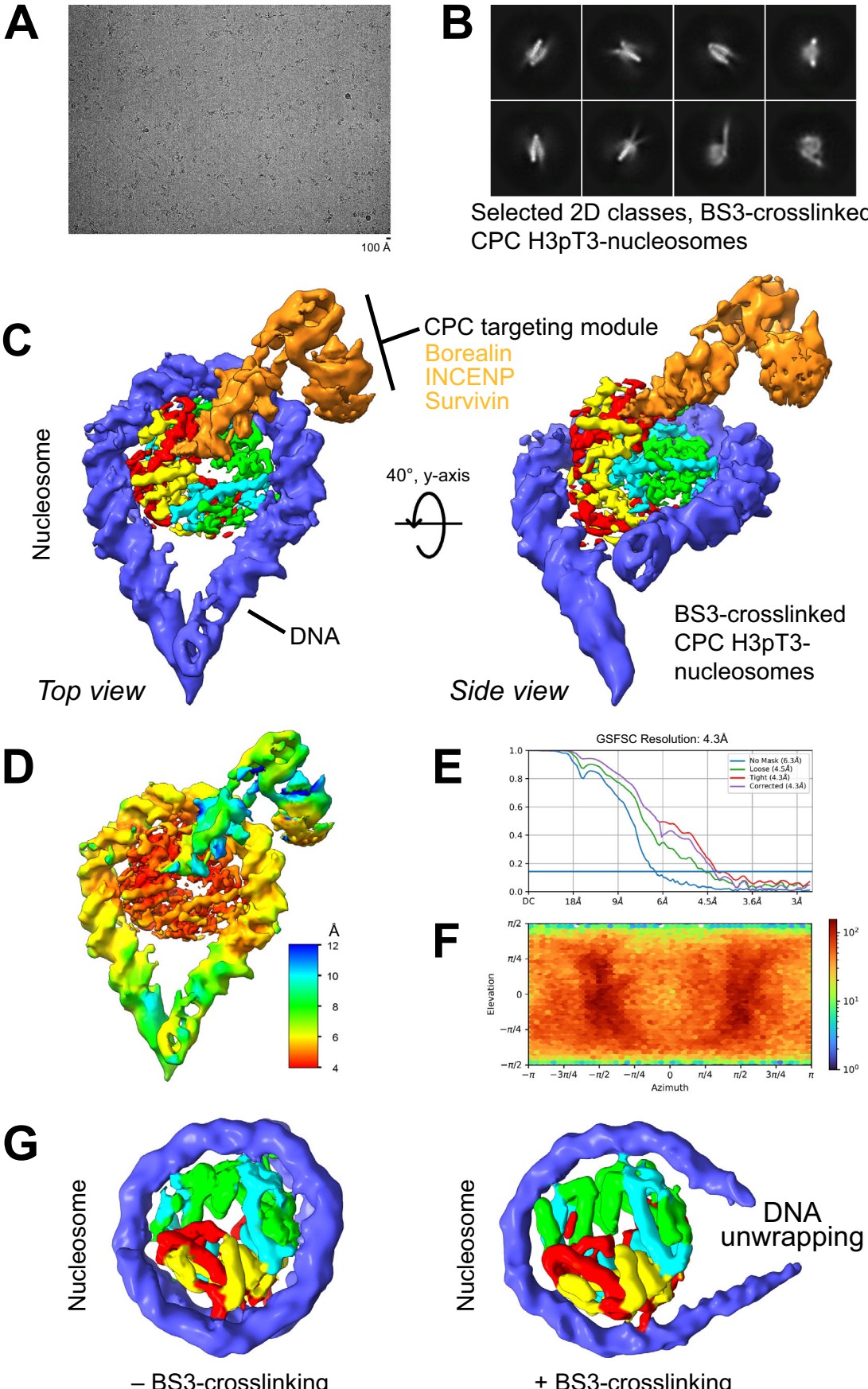

**A**

100 Å

**B**

Selected 2D classes, BS3-crosslinked
CPC H3pT3-nucleosomes

**C**

CPC targeting module
Borealin
INCENP
Survivin

Nucleosome

40°, y-axis

DNA

*Top view*

*Side view*

BS3-crosslinked
CPC H3pT3-
nucleosomes

**D**

Å
12
10
8
6
4

**E**

GSFSC Resolution: 4.3Å

No Mask (6.3Å)
Loose (4.5Å)
Tight (4.3Å)
Corrected (4.3Å)

DC   18Å   9Å   6Å   4.5Å   3.6Å   3Å

**F**

Elevation

Azimuth

**G**

Nucleosome

Nucleosome

DNA
unwrapping

– BS3-crosslinking

+ BS3-crosslinking

◄ **Figure EV3. BS3-crosslinked H3pT3-nucleosomes in complex with CPC80-Bor(FL).**

(A) Example micrograph showing BS3-crosslinked CPC80-Bor(FL) H3pT3-nucleosome particles. (B) Selected BS3-crosslinked 2D classes. (C) Final volume for a CPC80-Bor(FL) H3pT3-nucleosome showing histone H2A (red), H2B (yellow), H3 (green), H4 (cyan), DNA (blue), and CPC subunit density (orange) in top and side view. (D) A local resolution map for the CPC80-Bor(FL) H3pT3-nucleosome complex. (E) Gold-standard Fourier shell correlation resolution (GSFSC), blue horizontal line denotes FSC value of 0.143, and (F) particle orientation distribution in the final map. (G) Models for H3pT3 nucleosomes analysed without (–) or with (+) BS3-crosslinking.

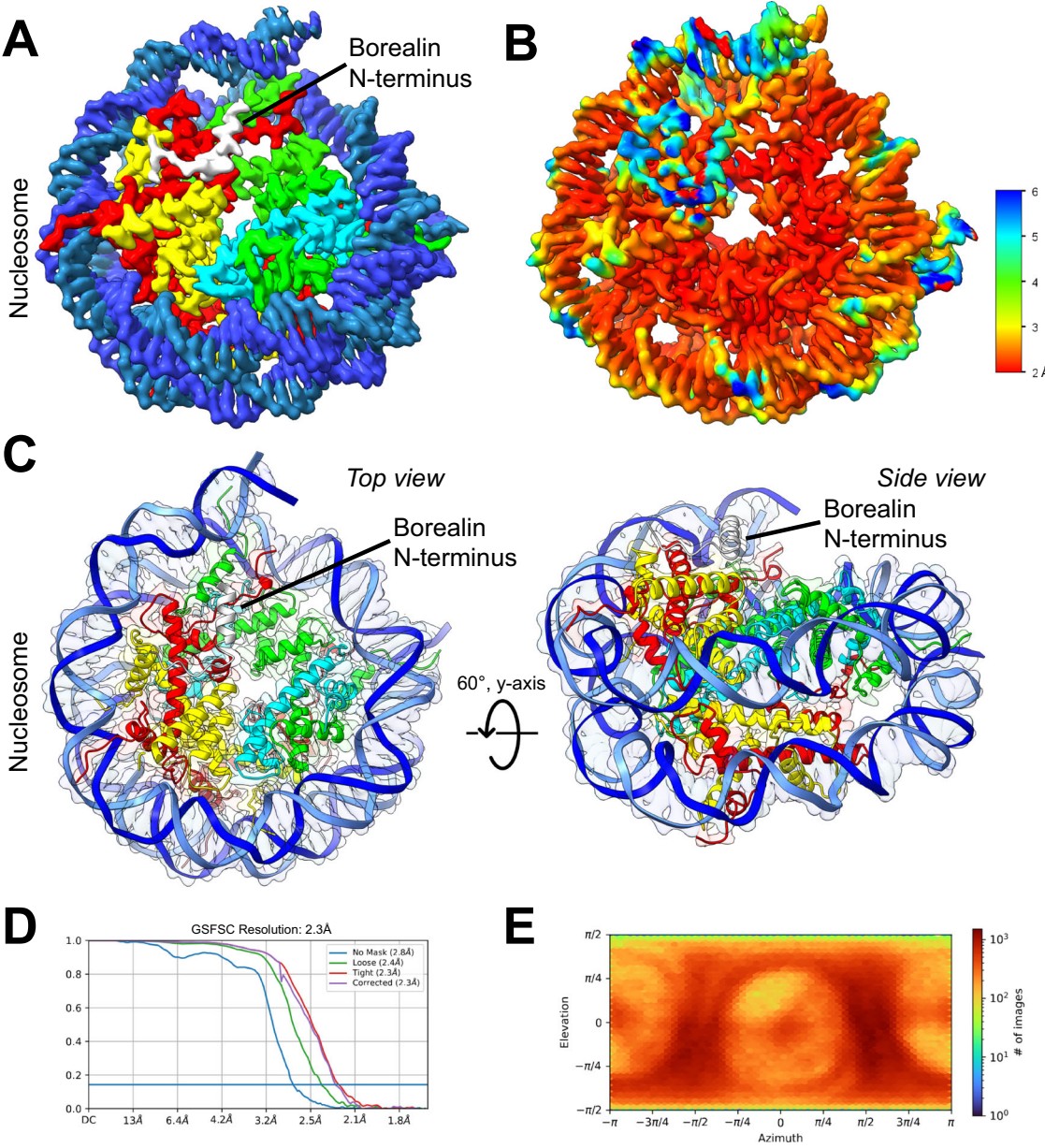

**Figure EV4. Interaction of borealin with the nucleosome acidic patch.**

(A) Cryo-EM density for the borealin N-terminus bound to a nucleosome showing histone H2A (red), H2B (yellow), H3 (green), H4 (cyan), DNA (blue), and borealin (grey/white), with (B) a colour-coded local resolution map (see inset scale 2–6 Å). PDB ID 8RUQ, EMDB ID - EMD-19514, Table EV1A (all particles). (C) Top (left) and side (right) views of the fitted model are shown. (D) Gold standard Fourier shell correlation (GSFSC) resolution, blue horizontal line indicates an FSC value of 0.143, and (E) particle orientation distribution within the final map.

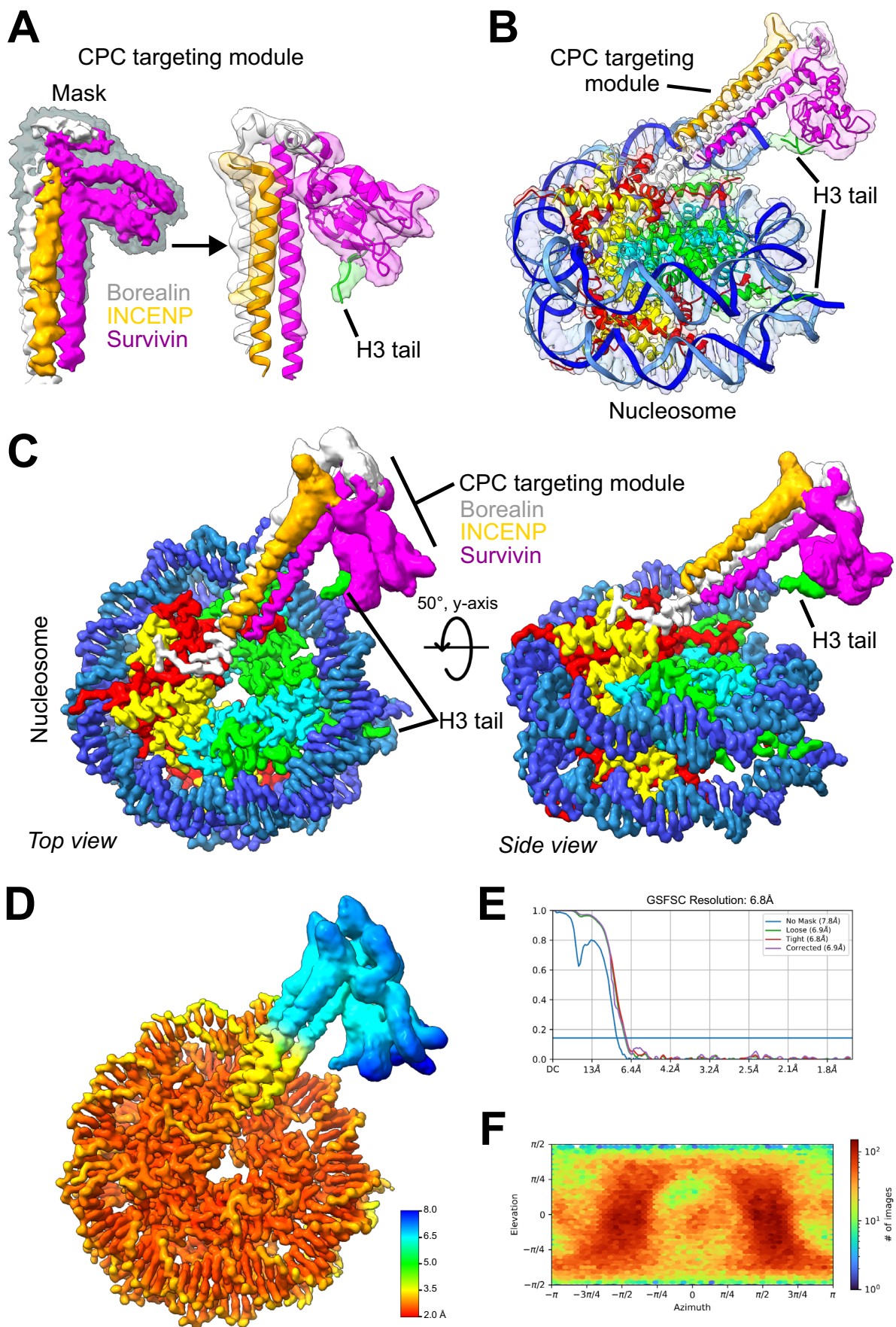

**Figure EV5.   Interaction of survivin with the tail of histone H3.**

(A) Local refinement of the CPC distal end with the masked region shaded in grey, with borealin (grey/white), INCENP (gold/orange), survivin (magenta), and histone H3 N-terminus (green). EMD-19684 and Table EV1C. (B) A fitted model for the CPC targeting module in complex with the H3pT3-nucleosome including H3 tail density associated with the survivin BIR domain. (C) Cryo-EM density of the CPC targeting module in complex with the H3pT3-nucleosome showing histone H2A (red), H2B (yellow), H3 (green); H4 (cyan), and CPC subunits borealin (grey/white), INCENP (gold/orange), and survivin (magenta). PDB ID 8RUP, EMDB ID–EMD-19513 and, Table EV1D (final). (D) A Local resolution map for the CPC targeting module in complex with the H3pT3-nucleosome including H3 tail density (green) associated with the survivin BIR domain. (E) Gold standard Fourier shell correlation (GSFSC) resolution, blue horizontal line indicates an FSC value of 0.143, and (F) particle orientation distribution within the final map.

