## [Peer Review File · EMBO Reports]

A pivot-tether model for nucleosome recognition by the chromosomal passenger complex

Reinis Ruza, Chyi Chung, Danny Gold, Michela Serena, Emile Roberts, Ulrike Gruneberg, and Francis Barr

Corresponding author(s): Francis Barr (francis.barr@bioch.ox.ac.uk)

Review Timeline:

Submission Date:	10th Feb 25
Editorial Decision:	17th Mar 25
Revision Received:	16th May 25
Editorial Decision:	20th Jun 25
Revision Received:	24th Jun 25
Accepted:	27th Jun 25

Editor: Deniz Senyilmaz Tiebe

Transaction Report:

Dear Francis,

Thank you for submitting your research manuscript to our journal, which was now seen by three referees, whose reports are copied below.

Referees express interest in the proposed model for nucleosome recognition by the chromosomal passenger complex. However, they also raise some concerns that need to be addressed to consider publication here. In particular,

- Referee #1 finds that the data currently does not differentiate between the clamp and bridge models, which need to be further tested (comment 1)
- Referee #3 requests further validation of the truncated borealin used in the study (1-76 aa).

Given these positive recommendations, we would like to invite you to submit a revised manuscript. Please revise your manuscript with the understanding that the referee concerns (as in their reports) must be fully addressed and their suggestions taken on board. Please address all referee concerns in a complete point-by-point response. Acceptance of the manuscript will depend on a positive outcome of a second round of review. It is EMBO reports policy to allow a single round of major experimental revision only and acceptance or rejection of the manuscript will therefore depend on the completeness of your responses included in the next, final version of the manuscript.

We realize that it is difficult to revise to a specific deadline. In the interest of protecting the conceptual advance provided by the work, we recommend a revision within 3 months. Please discuss the revision progress ahead of this time with me if you require more time to complete the revisions, or if you have questions or comments regarding the revision (also by video chat).

1. A data availability section providing access to data deposited in public databases is missing (where applicable).
2. Your manuscript contains statistics and error bars based on $n=2$. Please use scatter plots in these cases.

You can submit the revision either as a Scientific Report or as a Research Article. For Scientific Reports, the revised manuscript can contain up to 5 main figures and 5 Expanded View figures, and it should not exceed 27000 characters. If the revision leads to a manuscript with more than 5 main figures it will be published as a Research Article. In this case the Results and Discussion section should be separate. If a Scientific Report is submitted, these sections have to be combined. This will help to shorten the manuscript text by eliminating some redundancy that is inevitable when discussing the same experiments twice. In either case, all materials and methods should be included in the main manuscript file.

4) a .docx formatted letter INCLUDING the reviewers' reports and your detailed point-by-point responses to their comments. As part of the EMBO publication's Transparent Editorial Process, EMBO reports publishes online a Review Process File (RPF) to accompany accepted manuscripts. This File will be published in conjunction with your paper and will include the referee reports, your point-by-point response and all pertinent correspondence relating to the manuscript.

<https://www.embopress.org/page/journal/14693178/authorguide#transparentprocess>

5) a complete author checklist, which you can download from our author guidelines

<https://www.embopress.org/page/journal/14693178/authorguide>. Please insert information in the checklist that is also reflected in the manuscript. The completed author checklist will also be part of the RPF.

6) Please note that all corresponding authors are required to supply an ORCID ID for their name upon submission of a revised manuscript (). Please find instructions on how to link your ORCID ID to your account in our manuscript tracking system in our Author guidelines

7) Before submitting your revision, primary datasets produced in this study need to be deposited in an appropriate public database (see <https://www.embopress.org/page/journal/14693178/authorguide#datadeposition>). Please remember to provide a reviewer password if the datasets are not yet public. The accession numbers and database should be listed in a formal "Data Availability" section placed after Materials & Method (see also

<https://www.embopress.org/page/journal/14693178/authorguide#datadeposition>). Please note that the Data Availability Section is restricted to new primary data that are part of this study. * Note - All links should resolve to a page where the data can be accessed. *

Additional information on source data and instruction on how to label the files are available:

<https://www.embopress.org/page/journal/14693178/authorguide#sourcedata>

9) Our journal encourages inclusion of *data citations in the reference list* to directly cite datasets that were re-used and obtained from public databases. Data citations in the article text are distinct from normal bibliographical citations and should directly link to the database records from which the data can be accessed. In the main text, data citations are formatted as follows: "Data ref: Smith et al, 2001" or "Data ref: NCBI Sequence Read Archive PRJNA342805, 2017". In the Reference list, data citations must be labeled with "[DATASET]". A data reference must provide the database name, accession number/identifiers and a resolvable link to the landing page from which the data can be accessed at the end of the reference. Further instructions are available at <http://www.embopress.org/page/journal/14693178/authorguide#referencesformat>

10) Regarding data quantification (see Figure Legends:

<https://www.embopress.org/page/journal/14693178/authorguide#figureformat>)

12) Please also note our reference format:

13) All Materials and Methods need to be described in the main text using our 'Structured Methods' format, which is required for all research articles. According to this format, the Methods section includes a Reagents and Tools Table (listing key reagents, experimental models, software and relevant equipment and including their sources and relevant identifiers) followed by a Methods and Protocols section describing the methods using a step-by-step protocol format. The aim is to facilitate adoption of the methodologies across labs. More information on how to adhere to this format as well as a downloadable template (.docx) for the Reagents and Tools Table can be found in our author guidelines:

I look forward to seeing a revised version of your manuscript when it is ready. Please let me know if you have questions or comments regarding the revision.

Kind regards,

Deniz

Deniz Senyilmaz Tiebe, PhD
Senior Scientific Editor
EMBO Reports

Referee #1:

The manuscript by Ruza et al reports the cryo-EM structure of the nucleosome targeting module of the chromosome passenger complex, explaining how the CPC interacts with nucleosomes phosphorylated on Thr3 of histone H3, a modification catalysed by Haspin kinase. In addition to CPC targeting to pThr-H3 nucleosomes, the CPC is also targeted to histones phosphorylated on Thr120 of H2A - mediated by BUB1 kinase.

The CPC plays important roles to regulate the error correction mechanism at mitosis, ensuring sister chromatid pairs achieve bioriented attachment to the mitotic spindle and are under tension prior to anaphase onset. How the CPC senses tension and how lack of tension promotes CPC-mediated phosphorylation of the outer kinetochore to regulate the error correction mechanism is not fully understood. In one model, this process is regulated by the spatial separation between the inner centromere and inner kinetochore, where CPC is localised at metaphase, and the outer kinetochore targets of the CPC. In another model (not necessarily mutually exclusive) the CPC is sensitive to conformational changes in the outer kinetochore substrates induced by changes in tension. Thus understanding the mechanism of CPC interacts with the inner centromere/kinetochore is of considerable interest.

CPC interacts with pThr3(H3) through its BIR domain at the N-terminus of the survivin subunit, whereas the mechanism of targeting CPC to pThr120(H2A) nucleosomes, through Sgo1, is less well established. Additionally, recent studies showed that the N-terminus of the borealin subunit interacts with the acidic patch on H2A/H2B.

In this study the authors reconstituted a complex of the pThr3(H3)-nucleosome targeting module of the CPC (comprising surviving, borealin and the N-terminus of INCENP (residues 1-80) lacking the kinase domain) with a pThr3(H3)-nucleosome, thus allowing them to explain how the CPC recognises pThr3(H3)-nucleosomes. An initial cryo-EM reconstruction of this module with a pThr3(H3)-nucleosome with the CPC module and full length borealin generated a heterogeneous complex with partial DNA unwrapping from the histone octamer - likely caused by the BS3 cross-linker. In a revised reconstitution, the authors deleted the borealin dimerisation domain. The resulting complex generated a high resolution (2.3 Å) reconstruction. The structure indicates how the CPC module specifically recognises pT3H3 nucleosomes, showing how basic residues at the N-term of borealin interact with the acidic patch on H2A/H2B. The function of some of these basic residues were tested by mutagenesis and EMSA assays, and by monitoring CPC localisation in cells, showing reduced localisation. The authors observe cryo-EM density corresponding to the pThr3 segment of H3 interacting with the BIR1 domain of survivin. The link to the HFD of H3 is not present due to flexibility, however, from modelling constraints, the authors suggest it is the proximal H3 subunit that interacts with the CPC.

What is not observed in this structure are: (i) the interaction of the C-terminal loop of borealin with DNA and (ii) the interaction of the RRKKRR motif at the N-terminus of INCENP with DNA.

The finding that BS3 causes unwrapping of the DNA from the histone octamer is of interest and a valuable addition to the study.

Using STED microscopy the authors show that CPC is localised to the pericentromere and becomes separated from Ndc80 and CENP-A on transition from unaligned to aligned chromosomes.

Overall this is an interesting study that contributes to an understanding of the mechanism of CPC recruitment to pT3-H3 nucleosomes. The structural biology is in general of high quality and the figures are mainly clear. Specific points need to be addressed before acceptance.

Specific comments and questions

1. The authors propose that the dimeric CPC module bridges nucleosomes, rather than binding to a single nucleosome via a clamp. This claim appears to be based on their finding that dimer CPC (but not monomeric CPC) causes a supershift of nucleosomes preventing a complex of dimeric CPC and nucleosomes entering EMSA gels. Additionally the cryo-EM structure shows one CPC bound to a single nucleosome. This latter result does not distinguish between the clamp and bridge model, since in both models, one would expect two CPC modules/nucleosome. Lack of entry of a complex into an EMSA gel could be due to numerous reasons. The authors need to more vigorously test their model by defining the size of the CPC-nucleosome complex, for example mass photometry, SEC-MALS, AUC.
2. In the cryo-EM reconstruction of the modified CPC module with pT3-H3 nucleosome, the authors observe one CPC module bound to the nucleosome, whereas two would be expected. Did the authors determine the mol mass of the CPC-pT3-H3 nucleosome complex? They should do so if not, and based on these results comment further on the stoichiometry of the observed cryo-EM reconstruction of the CPC-pT3-H3 nucleosome complex.
3. Line 180: 'a single CPC binds to...' Do the authors mean a single CPC monomer or a single CPC dimer?
4. Line 219: The authors state a global resolution of 2.29 Å (Fig. S4). It isn't clear where this value comes from. The figure shows 2.42 Å for the unmasked map. Is 2.29 Å for the nucleosome with tight masking without CPC? Also use 2.3 Å (I doubt the accuracy of the 2.29 Å).
5. The localised 3D classification of the CPC resulted in a medium resolution reconstruction (i.e. around 6-7 Å). This should be stated explicitly in the results section.
6. Lines 269-271: 'Going from the H3pT3 nucleosome-proximal to the distal end of the CPC, the local resolution gradually got worse, with the survivin BIR domain being the most poorly defined region of the map (Fig. S4).' Fig. S4 does not show this. Do they mean Fig. S6?
7. Lines 139-159. It isn't clear how the data in this section addresses the question of whether the CPC controls error correction and the SAC by spatial separation or kinase activity.
8. The author used STED microscopy to visualise CPC and kinetochores at centromeres. Do they observe evidence for bi-partite centromeres?
9. Fig. 3B: Label residues.
10. In Figs 2, 4, S5, S6, rotation arrows are shown. The rotation angle needs to be cited. In Fig. 4A, E the rotation is shown about a single axis, i.e. vertical and horizontal. The actual rotation is around more than a single axis and this needs to be shown.
11. The methods are comprehensive. The exact sequences of CPC subunits cloned should be indicated, including the sequence of the His6 tag, protease site and intervening linkers.

Referee #2:

The paper by Ruza et al describe the cryoEM structure of one of the chromatin targeting motifs of the Chromosomal passenger complex (CPC) on a nucleosome. Specifically, the triple helical domain containing the N-terminal regions of Borealin and INCENP and the full length Survivin subunits bound to a nucleosome with a phosphorylated H3-T3 tail. The data nicely show how the N-terminus of borealin interact with the acidic patch of the nucleosome, while simultaneously the survivin bir domain interacts with the phosphorylated tail. This bipartite binding involving two unstructured regions allows the binding domain to "swing" which the authors postulate allows more productive collisions between the proteins.

In addition, the paper shows the higher resolution images of IF of the CPC with mitotic chromosomes than have been previously shown. Images employing a HCT116 cell line engineered to express a Halo-tagged Ndc80 protein and STED images provide evidence of overlap of Aurora kinase and kinetochores under prometaphase conditions and reduction of this overlap after microtubule attachments. While this is an old model in the literature the fact that this holds up with modern approaches is relevant. They reexamine the role of CPC recruitment of the MPS1 protein to kinetochores which is a key step in initiating the spindle checkpoint. Employing a new HeLa line, engineered to have MPS1-tagged with GFP they show that the CPC is required for MPS1 recruitment. This is similar, but not identical, to what has been published before using an MPS1 biosensor where there is a kinetic delay in MPS1 recruitment after the addition of Aurora B kinase inhibitors (PMID: 32888483). Finally they provide in vitro evidence that the CPC can bridge nucleosomes to form oligomeric nucleosomal complexes from mononucleosomes. Overall, I am enthusiastic about the work as the structure provides important new insight into the molecular interaction between a central mitotic regulator and chromatin, the ability of the interaction to swivel is surprising and interesting. Finally, IF data is of interest in that it shows that important models for CPC function cannot be discounted using higher resolution imaging technologies.

Specific Concerns:

Figure 1

- need westerns to show the degree of knockdown in siRNA experiments. Erich Nigg showed that the knock down of 1 CPC subunit led to knockdown of other subunits so please show all of the subunits.
- Gert Kops showed that Aurora B inhibition caused a delay but not complete loss of MPS1 at kinetochores (PMID: 21587233). It is important to test whether the differences between the experiments is your use of the MPS1-GFP where the tag could sensitize the localization. Do you see the same degree of loss of MPS1 as assayed by IF in your line of HeLa cells? If not, then the writing could be changed to highlight that the MPS1-GFP is a sensitized background to measure the role of CPC in MPS1 recruitment.
- The Figure legend specifies "Spindle checkpoint activity" but you measure MPS1 recruitment. A very small amount of MPS1 recruitment could generate the SAC so either directly measure the SAC activity or change the legend to MPS1 recruitment.
- Would borealin tail interactions with the acidic patch of the histone preclude Sgo1 binding PhosphoT120? This would predict that they would need to interact with different histone faces and would be worth mentioning, perhaps in the discussion.

Referee #3:

In this study the authors have investigated the localisation of the chromosome passenger complex by super-resolution microscopy and then identified the means by which it binds to nucleosomes by cryoEM. The authors first confirm previous studies showing that pericentromeric CPC is required for the Spindle Assembly Checkpoint. They subsequently use the Haspin kinase to phosphorylate nucleosomes on histone H3 threonine 3, which they bind to a partial CPC complex consisting of Aurora B, survivin, the first 76 amino acids of borealin and the first 80 amino acids of INCENP. The authors solve this structure using cryoEM and find that the N-terminus of borealis binds to the nucleosome acidic patch and acts as a pivot. The majority of the N-terminal tail of H3 is undefined in the structure, indicating that it is flexible, but the authors were able to resolve the T3-phosphorylated region binding to the BIR domain of surviving, in agreement with previous structures.

I have no issue with the super-resolution data, the biochemistry, or the cryoEM structures in this study. My concern, and it is a major concern, is that the structure uses truncated borealin but presents no data to validate their choice of the first 76 amino acids. How do we know that full-length borealis does not contain important binding sites that alter the behaviour of the CPC in vivo. Indeed, the authors present data in this study that a complex with full-length borealin plus cross-linking causes partial unwrapping of the DNA from nucleosomes. The authors dismiss this as a cross-linking artefact but it is possible that full length borealin has additional properties that could modulate its interaction with the nucleosome. Thus, I think it important that the authors validate their selection of the CPC with borealin 1-76 as a genuine surrogate for the CPC since their conclusion is that the CPC binds to nucleosomes through a borealin pivot.

Response to reviewers (replies in black text).**Referee #1:**

The manuscript by Ruza et al reports the cryo-EM structure of the nucleosome targeting module of the chromosome passenger complex, explaining how the CPC interacts with nucleosomes phosphorylated on Thr3 of histone H3, a modification catalysed by Haspin kinase. In addition to CPC targeting to pThr-H3 nucleosomes, the CPC is also targeted to histones phosphorylated on Thr120 of H2A - mediated by BUB1 kinase.

The CPC plays important roles to regulate the error correction mechanism at mitosis, ensuring sister chromatid pairs achieve bioriented attachment to the mitotic spindle and are under tension prior to anaphase onset. How the CPC senses tension and how lack of tension promotes CPC-mediated phosphorylation of the outer kinetochore to regulate the error correction mechanism is not fully understood. In one model, this process is regulated by the spatial separation between the inner centromere and inner kinetochore, where CPC is localised at metaphase, and the outer kinetochore targets of the CPC. In another model (not necessarily mutually exclusive) the CPC is sensitive to conformational changes in the outer kinetochore substrates induced by changes in tension. Thus, understanding the mechanism of CPC interacts with the inner centromere/kinetochore is of considerable interest.

CPC interacts with pThr3(H3) through its BIR domain at the N-terminus of the survivin subunit, whereas the mechanism of targeting CPC to pThr120(H2A) nucleosomes, through Sgol, is less well established. Additionally, recent studies showed that the N-terminus of the borealin subunit interacts with the acidic patch on H2A/H2B.

In this study the authors reconstituted a complex of the pThr3(H3)-nucleosome targeting module of the CPC (comprising survivin, borealin and the N-terminus of INCENP (residues 1-80) lacking the kinase domain) with a pThr3(H3)-nucleosome, thus allowing them to explain how the CPC recognises pThr3(H3)-nucleosomes. An initial cryo-EM reconstruction of this module with a pThr3(H3)-nucleosome with the CPC module and full length borealin generated a heterogeneous complex with partial DNA unwrapping from the histone octamer - likely caused by the BS3 cross-linker. In a revised reconstitution, the authors deleted the borealin dimerisation domain. The resulting complex generated a high resolution (2.3 Å) reconstruction. The structure indicates how the CPC module specifically recognises pT3H3 nucleosomes, showing how basic residues at the N-term of borealin interact with the acidic patch on H2A/H2B. The function of some of these basic residues were tested by mutagenesis and EMSA assays, and by monitoring CPC localisation in cells, showing reduced localisation. The authors observe cryo-EM density corresponding to the pThr3 segment of H3 interacting with the BIR1 domain of survivin. The link to the HFD of H3 is not present due to flexibility, however, from modelling constraints, the authors suggest it is the proximal H3 subunit that interacts with the CPC.

What is not observed in this structure are: (i) the interaction of the C-terminal loop of borealin with DNA and (ii) the interaction of the RRKKRR motif at the N-terminus of INCENP with DNA.

The finding that BS3 causes unwrapping of the DNA from the histone octamer is of interest and a valuable addition to the study.

Using STED microscopy the authors show that CPC is localised to the pericentromere and becomes separated from Ndc80 and CENP-A on transition from unaligned to aligned chromosomes.

Overall this is an interesting study that contributes to an understanding of the mechanism of CPC recruitment to pT3-H3 nucleosomes. The structural biology is in general of high quality and the figures are mainly clear. Specific points need to be addressed before acceptance.

Specific comments and questions

1. The authors propose that the dimeric CPC module bridges nucleosomes, rather than binding to a single nucleosome via a clamp. This claim appears to be based on their finding that dimer CPC (but not monomeric CPC) causes a supershift of nucleosomes preventing a complex of dimeric CPC and nucleosomes entering EMSA gels. Additionally, the cryo-EM structure shows one CPC bound to a single nucleosome. This latter result does not distinguish between the clamp and bridge model, since in both models, one would expect two CPC modules/nucleosome. Lack of entry of a complex into an EMSA gel could be due to numerous reasons. The authors need to more vigorously test their model by defining the size of the CPC-nucleosome complex, for example mass photometry, SEC-MALS, AUC.

Response: We carried out extensive experiments with different molar ratios of CPC:nucleosomes trying to coelute both components on size-exclusion columns, going up to a 32x excess of CPC, but were unsuccessful with that approach. The main reason for this outcome is the apparently low affinity between H3pT3-nucleosomes and the CPC/Bor76 complex. However, we were able to visualise complex formation using EMSA assays which are shown in the manuscript. For the preparation of the final sample used for structure determination we used up to 7x excess of CPC, which was the practical highest before CPC particles started to take over the EM grids and it became very difficult to distinguish the nucleosome particles present. Based on the observations supported by

EMSA data, our reasoning was that using this sample for determining the size of the complex would not lead to any detection of the CPC-bound to nucleosomes, because either they would get separated out (SEC-MALS, AUC), or because the duration of the transient interaction would be too short for detection by mass photometry. In summary, it seems like the affinity of the CPC construct used was too low to obtain a complex with two CPCs bound that would be stable enough for detection by any of the biophysical methods proposed. In addition, even if these methods were used, the observation of either CPC-unbound or single-CPC bound nucleosomes would not serve as conclusive evidence that the two CPC bound form does not exist *in vivo* due to the low binding affinity of the CPC complexes used. Further work is needed to determine whether CPC can bridge nucleosomes in the context of chromatin, and this requires other approaches (e.g. cryo-ET) which are outside the scope of this study.

2. In the cryo-EM reconstruction of the modified CPC module with pT3-H3 nucleosome, the authors observe one CPC module bound to the nucleosome, whereas two would be expected. Did the authors determine the mol mass of the CPC-pT3-H3 nucleosome complex? They should do so if not and based on these results comment further on the stoichiometry of the observed cryo-EM reconstruction of the CPC-pT3-H3 nucleosome complex.

Response: As explained in the response to Q1 it was not possible to make precise mass determinations of the CPC module with the H3pT3 nucleosomes due to technical challenges. It is therefore difficult to comment further on the stoichiometry from the cryo-EM structures and we would prefer to keep with our original text and conclusions on this point. It is important to note that nucleosomes are present at a concentration of ~80-100 μm in the nucleus of eukaryotic cells and chromatin binding complexes therefore do not typically show high affinity binding. This poses technical challenges and complicates biophysical analysis of the sort proposed. As we show, methods to stabilise complexes such as crosslinking simply introduce unwanted changes of their own and are potentially misleading. However, we do not think these limitations reduce the value or importance of our work. For the Aurora B and the CPC which undergo dynamic relocation from pericentromeres to the central spindle at the metaphase to anaphase transition this relatively low binding affinity is probably crucial to prevent Aurora B trapping on chromatin.

3. Line 180: 'a single CPC binds to...' Do the authors mean a single CPC monomer or a single CPC dimer?

Response: Revised text on lines 193-195 clarifies this point: "*Based on the biochemical data we favour the idea that dimeric CPC binds to a single face on each nucleosome, and CPC bridges between nucleosomes rather than acting as a clamp capturing a single nucleosome.*"

4. Line 219: The authors state a global resolution of 2.29 Å (Fig. S4). It isn't clear where this value comes from. The figure shows 2.42 Å for the unmasked map. Is 2.29 Å for the nucleosome with tight masking without CPC? Also use 2.3 Å (I doubt the accuracy of the 2.29 Å).

Response: Yes, 2.29Å is for the tightly masked model (with only Borealin N-terminus), 2.42Å is for the unmasked CPC map (before local refinements). These figures have been rounded to 2.3Å and 2.4Å in the revised text and figures.

5. The localised 3D classification of the CPC resulted in a medium resolution reconstruction (i.e. around 6-7 Å). This should be stated explicitly in the results section.

Response: This information was included in the cryo-EM data processing figure (now Fig. S5) and is now explicitly stated on line 283 of the manuscript.: "*To explore this idea, localised 3D-classification focusing on the CPC rather than the H3pT3 nucleosome density was performed to give a medium resolution reconstruction of 6-7Å.*"

6. Lines 269-271: 'Going from the H3pT3 nucleosome-proximal to the distal end of the CPC, the local resolution gradually got worse, with the survivin BIR domain being the most poorly defined region of the map (Fig. S4).' Fig. S4 does not show this. Do they mean Fig. S6?

Response: This error has been corrected in the revised manuscript. Lines 295-297: "*Going from the H3pT3 nucleosome-proximal to the distal end of the CPC, the local resolution gradually worsened, with the survivin BIR domain being the most poorly defined region of the map (Fig. EV5).*"

7. Lines 139-159. It isn't clear how the data in this section addresses the question of whether the CPC controls error correction and the SAC by spatial separation or kinase activity.

Response: Depletion of any of the CPC subunits results in both re-localisation of the CPC to the cytoplasm (revised Figure 2) as well as a drop in the levels of Aurora B kinase levels and related kinase activity. This is now made clear in the text. In the new revised Figure 2 we show Aurora B localisation in the CPC subunit depletion experiment, demonstrating complete loss of centromere and chromatin targeting of Aurora B when individual CPC subunits are depleted. We have also included Western blots demonstrating reduced Aurora B total levels upon CPC subunit depletion, as well as immunofluorescence analysis demonstrating depletion of INCENP, survivin and borealin in the supplement (Figure S2C and 2D). The experiments presented in Figure 2 therefore

cannot clearly separate the effects of loss of CPC localisation and impairment of CPC activity on MPS1 localisation. The survivin RNAi experiment presented in Figure 5C and 5D, however, demonstrates that loss of CPC pericentromeric localisation in the presence of near-normal activity makes a substantial difference to MPS1 kinetochore recruitment, arguing that spatial control of the CPC is critical for spindle checkpoint activation. This has now been more clearly explained in the discussion text (lines 337-348).

8. The author used STED microscopy to visualise CPC and kinetochores at centromeres. Do they observe evidence for bi-partite centromeres?

Response: We have not observed bi-partite structures with different CPC subunits or kinetochore proteins. However, Sgo1 does show bi-partite staining which we have briefly discussed (lines 138-139, and 414-418).

9. Fig. 3B: Label residues.

Response: Residues are labelled in the revised figure (Figure 4B) in addition to the labelling already present in Figure 4C.

10. In Figs 2, 4, S5, S6, rotation arrows are shown. The rotation angle needs to be cited. In Fig. 4A, E the rotation is shown about a single axis, i.e. vertical and horizontal. The actual rotation is around more than a single axis and this needs to be shown.

Response: Rotations are as follows (note new figure numbering):

- Fig. 3A: 10° x-axis (horizontal), 55° y-axis (vertical)
- Fig. 5A: 170° z (frontal), 110° y-axis
- Fig. 5B: 180° x, 30° y-axis
- EV Fig 3C: 40° y-axis
- EV Fig4C: 60° y-axis
- EV Fig 5C: 50° y-axis

This information has been added to the revised figures as requested.

11. The methods are comprehensive. The exact sequences of CPC subunits cloned should be indicated, including the sequence of the His6 tag, protease site and intervening linkers.

Response: The vectors and regions of human CPC subunits used are described in the methods section on lines 640-665. The protease cleavage sites, linker and tags are the standard sequences for these vectors as described in the text.

Referee #2:

The paper by Ruza et al describe the cryoEM structure of one of the chromatin targeting motifs of the Chromosomal passenger complex (CPC) on a nucleosome. Specifically, the triple helical domain containing the N-terminal regions of Borealin and INCENP and the full length Survivin subunits bound to a nucleosome with a phosphorylated H3-T3 tail. The data nicely show how the N-terminus of borealin interact with the acidic patch of the nucleosome, while simultaneously the survivin bir domain interacts with the phosphorylated tail. This bipartite binding involving two unstructured regions allows the binding domain to "swing" which the authors postulate allows more productive collisions between the proteins.

In addition, the paper shows the higher resolution images of IF of the CPC with mitotic chromosomes than have been previously show. Images employing a HCT116 cell line engineered to express a Halo-tagged Ndc80 protein and STED images provide evidence of overlap of Aurora kinase and kinetochores under prometaphase conditions and reduction of this overlap after microtubule attachments. While this is an old model in the literature the fact that this holds up with modern approaches is relevant. They reexamine the role of CPC recruitment of the MPS1 protein to kinetochores which is a key step in initiating the spindle checkpoint. Employing a new HeLa line, engineered to have MPS1-tagged with GFP they show that the CPC is required for MPS1 recruitment. This is similar, but not identical, to what has been published before using an MPS1 biosensor where there is a kinetic delay in MPS1 recruitment after the addition of Aurora B kinase inhibitors (PMID: 32888483). Finally they provide in vitro evidence that the CPC can bridge nucleosomes to form oligomeric nucleosomal complexes from mononucleosomes.

Overall, I am enthusiastic about the work as the structure provides important new insight into the molecular interaction between a central mitotic regulator and chromatin, the ability of the interaction to swivel is surprising and interesting. Finally, IF data is of interest in that it shows that important models for CPC function cannot be discounted using higher resolution imaging technologies.

Specific Concerns:

Figure 1

-need westerns to show the degree of knockdown in siRNA experiments. Erich Nigg showed that the knock down of 1 CPC subunit led to knockdown of other subunits so please show all of the subunits.

Response: This data and relevant citations have been added to the manuscript (Figure S2) and revised text on lines 155-158).

-Gert Kops showed that Aurora B inhibition caused a delay but not complete loss of MPS1 at kinetochores (PMID: 21587233). It is important to test whether the differences between the experiments is your use of the MPS1-GFP where the tag could sensitize the localization. Do you see the same degree of loss of MPS1 as assayed by IF in your line of HeLa cells? If not, then the writing could be changed to highlight that the MPS1-GFP is a sensitized background to measure the role of CPC in MPS1 recruitment.

Response: MPS1 is difficult to detect by immunofluorescence analysis because of the poor signal to noise ratio seen with most available antibodies, most likely due to interference of antibody recognition by extensive mitotic auto-phosphorylation of MPS1. Most researchers therefore carry out antibody staining for MPS1 in the presence of MPS1 inhibitors, enabling better detection of the inhibited but not active kinase. The endogenously tagged CRISPR cells lines that we have used in this and several previous studies (Hayward et al., 2019; Alfonso-Perez et al., 2019; Hayward et al., 2019b; Hayward, Roberts and Gruneberg, 2022) circumvent this problem and offer much improved detection of the active MPS1 protein. The cells lines have been extensively tested and have normal mitotic timing and spindle checkpoint responses, and the fluorescent tag does not seem to interfere with the function of the molecule, so we do not think they are "sensitized" in that sense. The study by the Kops lab, that the referee refers to, which we confirm and extend, shows significantly reduced MPS1 kinetochore levels upon Aurora B inhibition, very similar to our study here. Small differences in the extent of MPS1 loss from kinetochores upon Aurora B inhibition are most likely attributable to the Aurora B inhibitor not being 100% effective under the conditions used in the Kops study rather than the use of different detection methods. In our experience it is often technically challenging to achieve complete Aurora B inhibition without affecting the related kinase Aurora A, a limitation applicable to all studies in this area.

-The Figure legend specifies "Spindle checkpoint activity" but you measure MPS1 recruitment. A very small amount of MPS1 recruitment could generate the SAC so either directly measure the SAC activity or change the legend to MPS1 recruitment.

Response: We have changed the titles for revised Figure 4 and Figure 5 to say "*MPS1 recruitment*" and provided data on BUBR1 in the new Figure 2 to show checkpoint activity downstream of MPS1.

-Would borealin tail interactions with the acidic patch of the histone preclude Sgo1 binding PhosphoT120? This would predict that they would need to interact with different histone faces and would be worth mentioning, perhaps in the discussion.

Response: We agree the referee that the idea different interactions occur on the two faces of the same nucleosomes is an interesting possibility. However, we lack data on Sgo1 and would therefore prefer not to

speculate on this point. We do suggest additional follow up studies should address Sgo1 and additional histone modifications in the discussion.

Referee #3:

In this study the authors have investigated the localisation of the chromosome passenger complex by super-resolution microscopy and then identified the means by which it binds to nucleosomes by cryoEM. The authors first confirm previous studies showing that pericentromeric CPC is required for the Spindle Assembly Checkpoint. They subsequently use the Haspin kinase to phosphorylate nucleosomes on histone H3 threonine 3, which they bind to a partial CPC complex consisting of Aurora B, survivin, the first 76 amino acids of borealin and the first 80 amino acids of INCENP. The authors solve this structure using cryoEM and find that the N-terminus of borealis binds to the nucleosome acidic patch and acts as a pivot. The majority of the N-terminal tail of H3 is undefined in the structure, indicating that it is flexible, but the authors were able to resolve the T3-phosphorylated region binding to the BIR domain of survivin, in agreement with previous structures.

I have no issue with the super-resolution data, the biochemistry, or the cryoEM structures in this study. My concern, and it is a major concern, is that the structure uses truncated borealin but presents no data to validate their choice of the first 76 amino acids. How do we know that full-length borealis does not contain important binding sites that alter the behaviour of the CPC in vivo. Indeed, the authors present data in this study that a complex with full-length borealin plus cross-linking causes partial unwrapping of the DNA from nucleosomes. The authors dismiss this as a cross-linking artefact, but it is possible that full length borealin has additional properties that could modulate its interaction with the nucleosome. Thus, I think it important that the authors validate their selection of the CPC with borealin 1-76 as a genuine surrogate for the CPC since their conclusion is that the CPC binds to nucleosomes through a borealin pivot.

Response: Prior to deciding on constructs for our structural biology analysis, we performed an analysis of borealin truncations and their ability to target to chromosomes in metaphase. This revealed that although full-length borealin localises to centromeres most efficiently, truncations to position 76 are still centromeric albeit with slightly increased cytoplasmic staining (now included as Figure S4). This supports our choice of borealin 1-76 for reconstitution into minimal CPC targeting module complexes for further detailed structural studies, in addition to the full-length borealin protein.

Figure S4. Borealin truncations define a minimal fragment mediating chromatin targeting. HeLa cells were transfected with the indicated borealin-GFP truncation constructs for 24 h and then fixed and processed for microscopy. Centromeres were stained with anti-centromere antibodies (predominantly CENP-A/B/C). DNA was detected using Hoechst-33528. Single images are shown representative of multiple independent experiments. Scale bar: 10 µm.

As we establish in our manuscript, the properties of complexes with full-length borealin made them unsuitable for structural studies due to their propensity to precipitate. This necessitated the use of BS3-crosslinking to stabilise the CPC-H3pT3 nucleosome particles on grids (described in Figure EV3). With these samples we observed DNA unwrapping from H3pT3 nucleosomes bound to CPC with full length borealin after BS3-crosslinking. The reviewer states: “The authors dismiss this as a cross-linking artefact but it is possible that full length borealin has additional properties that could modulate its interaction with the nucleosome.” Our data make the possibility raised by the referee unlikely. As shown in the original and revised figures, we have collected cryo-EM data with BS3-crosslinked nucleosomes without CPC and these samples also showed DNA unwrapping (Figure EV3G).

Therefore, it is not correct to write that we “dismiss” the possibility that full-length borealin might induce DNA unwrapping from nucleosomes, when we clearly show that DNA unwrapping can be caused by BS3-crosslinking of nucleosomes in the absence of any CPC. The simplest conclusion is that borealin plays no role in DNA unwrapping in these experiments. We also note that referee 1 accepted this idea and wrote that “*The finding that BS3 causes unwrapping of the DNA from the histone octamer is of interest and a valuable addition to the study.*”.

In summary, the use of a minimal CPC targeting module enables us to collect valuable data and to propose a model of how this might function *in vivo*, which is a common reductionist approach. This data supports the key conclusions that the CPC binds nucleosomes through a combined borealin pivot and H3pT3-tether anchored to survivin. We have not claimed that other regions of the CPC are irrelevant and indeed note in our discussion that further work is required to understand other important factors such as Shugoshin, other histone modifications and components of centromeres.

Dear Prof. Barr,

Thank you for submitting your revised manuscript. It has now been seen by two of the original referees. My apologies for the delay in getting back to you, which was due to the delay in receiving referee reports.

As you will see, referees find that the study is significantly improved during revision and recommend publication. However, referee #1 has remaining outstanding concerns. Please address them by making textual alterations. Please provide a point-by-point response outlining the textual alterations made in response to each comment.

Moreover, the editorial points below need to be addressed before I can accept the manuscript.

- Please provide 3-5 keywords for your study. These will be visible in the html version of the paper and on PubMed and will help increase the discoverability of your work.
- To conform to our format requirements, please rename the "Data and materials availability" section as "Data Availability" and remove the sentence "Reagents generated in this study can be obtained from the corresponding author."
- Relate to the point above, the specific URLs for S-BSST2028, PDB 8RUP, PDB ID 8RUQ, EMD-19513, EMD-19514, EMD-19685, and EMD-19684 need to be provided in the data availability statement.
- Please rename the "Competing interests" section as "Disclosure and Competing Interests Statement".
- Please remove the Author Contributions section from the manuscript text.
- We note that the cells B6 and D108 have currently not been filled in in the Author checklist.
- Regarding the Funding Information, we note that MRC PhD studentships have not been entered into the manuscript submission system. Funding section heading needs to be removed, information should be included in Acknowledgments.
- We note that the panels of Figure 3 have not been individually called out in the text.
- We note the following regarding the Appendix: A Table of Contents need to be added. The 5 Appendix figures are currently uploaded as separate files, these need to be provided in a single PDF file entitled Appendix and the correct nomenclature should be Appendix Figure S1-S5; their legends should be removed from the manuscript file and each should follow its figure in the Appendix PDF. A title page with a short Table of Contents need to be added with page numbers; the nomenclature should be corrected in all places: Appendix file, manuscript callouts.
- We note that movies were referred to in the text, but not provided.
- All research articles submitted as revised versions must include a structured methods section that includes a Reagents and Tools Table followed by a Methods and Protocols section. Please see <https://www.embopress.org/page/journal/14693178/authorguide#structuredmethods> for further information.
- Materials and Methods should be renamed as Methods.
- The manuscript sections should be in the following order: Title page - Abstract & Keywords - Introduction - Results - Discussion - Methods - Data Availability - Acknowledgments - Disclosure Statement & Competing Interests - References - Figure Legends - (Main Tables with legends if applicable) - Expanded View Figure Legends.
- Extended View Figures should be renamed as Expanded View Figure Legends.
- Please resubmit Table 1 as Table EV1 and update callouts and the file name accordingly.
- Our production/data editors have asked you to clarify several points in the figure legends - Figure Legends (main + EV):
 - o Please note that the exact p values are not provided in the legends of figures 1A, B; 2B, C, D, F, G; 4F, 5D, EV1, S2B
 - o Please note that scale bar and its definition are missing for figure EV3 A
- Papers published in EMBO Reports include a 'synopsis' and 'bullet points' to further enhance discoverability. Both are displayed on the html version of the paper and are freely accessible to all readers. The synopsis includes a short standfirst summarizing the study in 1 or 2 sentences (max 35 words) that summarize the paper and are provided by the authors and streamlined by the handling editor. I would therefore ask you to include your synopsis blurb and 3-5 bullet points listing the key experimental findings.
- In addition, please provide an image for the synopsis. This image should provide a rapid overview of the question addressed in the study but still needs to be kept fairly modest since the image size cannot exceed 550 (width) x 300-600 (height) pixels.

Thank you again for giving us to consider your manuscript for EMBO Reports, I look forward to your minor revision.

Kind regards,

Deniz Senyilmaz Tiebe

--
Deniz Senyilmaz Tiebe, PhD
Senior Scientific Editor
EMBO Reports

Referee #1:

The authors' response to the review is disappointing.

Point 1:

It is not clear that in the revised manuscript any effort has been made to address concerns about their proposal that the CPC dimer bridges adjacent nucleosomes rather than the alternative scenario that the CPC dimer binds a single nucleosome. The authors maintain that the EMSA result provides 'biochemical' evidence for the bridging model. It might be consistent with that, but other explanations for the inconclusive EMSA experiment are possible. The failure of the sample to enter an EMSA gel is not quantitative biochemical data. The authors should caveat their proposal. The sentence: 'Alternatively, as our biochemical data indicate, the CPC may bridge between nucleosomes to create chains in a manner dependent on borealin dimerization and interaction of survivin with the H3pT3 modification' needs to be modified: 'suggest' substituted for 'proposed'.

Additionally, the authors need to discuss what other experiments were conducted to test the bridge versus clamp model, and the results obtained, and mention that further work is required to determine whether CPC can bridge nucleosomes in the context of nucleosomes. This reviewer cannot recommend publication without these statements.

Referee #2:

The authors have adequately addressed my concerns and I find the revised manuscript suitable for publication in EMBO reports.

Senior Scientific Editor
EMBO Reports

Referee #1:

The authors' response to the review is disappointing.

Point 1:

It is not clear that in the revised manuscript any effort has been made to address concerns about their proposal that the CPC dimer bridges adjacent nucleosomes rather than the alternative scenario that the CPC dimer binds a single nucleosome. The authors maintain that the EMSA result provides 'biochemical' evidence for the bridging model. It might be consistent with that, but other explanations for the inconclusive EMSA experiment are possible. The failure of the sample to enter an EMSA gel is not quantitative biochemical data. The authors should caveat their proposal. The sentence: 'Alternatively, as our biochemical data indicate, the CPC may bridge between nucleosomes to create chains in a manner dependent on borealin dimerization and interaction of survivin with the H3pT3 modification' needs to be modified: 'suggest' substituted for 'proposed'.

In our revised submission we addressed 10 of the 11 points raised by the reviewer and explained why we did not show more analysis of the CPC bridging model. The focus of our work was the mechanism by which the CPC binds to nucleosomes and recruits Aurora B to promote spindle assembly checkpoint signalling. In doing that, we prepared and tested the properties of different CPC-nucleosome complexes for cryo-EM analysis or using EMSA assays and describe the latter data in Figure EV2. While nucleosomes bound by monovalent CPC complexes enter the EMSA gel, those bound by dimeric CPC complexes remain close to the top of the gel indicating that they are larger or have altered properties. Those results are not inconclusive, even if they have different possible explanations. We suggest that dimeric CPC complexes may bridge or link nucleosomes. However, we don't exclude other possibilities. Indeed, one possibility is that the CPC phase separates to collect nucleosomes into a molecular condensate as suggested in previous work which is mentioned in the introduction. The EMSA data is discussed in a measured fashion and is not mentioned in the Abstract or manuscript title which we feel was the right decision.

The referee asks for a change in the discussion text from "suggest" to "propose". This is confusing since we used the word "indicate" in a context where "propose" cannot be used. We believe the referee wishes "indicate" to be changed to "suggest" and we have done this. In addition, we have more clearly explained the limitations of our study and the need for future work – see below.

Additionally, the authors need to discuss what other experiments were conducted to test the bridge versus clamp model, and the results obtained, and mention that further work is required to determine whether CPC can bridge nucleosomes in the context of nucleosomes. This reviewer cannot recommend publication without these statements.

Revised text:

"Therefore, we cannot rule out that a clamp binding mode facilitated by borealin dimerization could exist in vivo. Alternatively, as our biochemical data suggest, the CPC may bridge between nucleosomes to create chains in a manner dependent on borealin dimerization and interaction of survivin with the H3pT3 modification (Fig. EV2A and EV2B). Thus, while our study provides important information about the interaction of the CPC with individual nucleosomes, further work will be important to address how the CPC interacts with nucleosomes in the context of chromatin in vivo and the potential for different clamping and bridging modes of interaction."

Referee #2:

The authors have adequately addressed my concerns and I find the revised manuscript suitable for publication in EMBO reports.

We thank the reviewer for their words and helpful suggestions during the review.

Prof. Francis Barr
University of Oxford
Department of Biochemistry
South Parks Road
Oxford OX1 3QU
United Kingdom

Dear Francis,

Thank you for submitting your revised manuscript. I have now looked at everything and all is fine. Therefore, I am very pleased to accept your manuscript for publication in EMBO Reports.

Congratulations on a nice work!

Kind regards,

Deniz

--

Deniz Senyilmaz Tiebe, PhD
Senior Scientific Editor
EMBO Reports

--
